# POLYNORMER: POLYNOMIAL-EXPRESSIVE GRAPH TRANSFORMER IN LINEAR TIME

**Chenhui Deng, Zichao Yue, Zhiru Zhang**
Cornell University, Ithaca, USA
`{cd574, zy383, zhiruz}@cornell.edu`

## ABSTRACT

Graph transformers (GTs) have emerged as a promising architecture that is theoretically more expressive than message-passing graph neural networks (GNNs). However, typical GT models have at least quadratic complexity and thus cannot scale to large graphs. While there are several linear GTs recently proposed, they still lag behind GNN counterparts on several popular graph datasets, which poses a critical concern on their practical expressivity. To balance the trade-off between expressivity and scalability of GTs, we propose Polynormer, a polynomial-expressive GT model with linear complexity. Polynormer is built upon a novel base model that learns a high-degree polynomial on input features. To enable the base model permutation equivariant, we integrate it with graph topology and node features separately, resulting in local and global equivariant attention models. Consequently, Polynormer adopts a linear local-to-global attention scheme to learn high-degree equivariant polynomials whose coefficients are controlled by attention scores. Polynormer has been evaluated on 13 homophilic and heterophilic datasets, including large graphs with millions of nodes. Our extensive experiment results show that Polynormer outperforms state-of-the-art GNN and GT baselines on most datasets, even without the use of nonlinear activation functions. Source code of Polynormer is freely available at: github.com/cornell-zhang/Polynormer.

## 1 INTRODUCTION

As conventional graph neural networks (GNNs) are built upon the message passing scheme by exchanging information between adjacent nodes, they are known to suffer from *over-smoothing* and *over-squashing* issues (Oono & Suzuki, 2020; Alon & Yahav, 2021; Di Giovanni et al., 2023), resulting in their limited expressive power to (approximately) represent complex functions (Xu et al., 2018; Oono & Suzuki, 2020). Inspired by the advancements of Transformer-based models in language and vision domains (Vaswani et al., 2017; Dosovitskiy et al., 2021), graph transformers (GTs) have become increasingly popular in recent years, which allow nodes to attend to all other nodes in a graph and inherently overcome the aforementioned limitations of GNNs. In particular, Kreuzer et al. (2021) have theoretically shown that GTs with unbounded layers are universal equivariant function approximators on graphs. However, it is still unclear how to unlock the expressivity potential of GTs in practice since the number of GT layers is typically restricted to a small constant.

In literature, several prior studies have attempted to enhance GT expressivity by properly involving inductive bias through positional encoding (PE) and structural encoding (SE). Specifically, Ying et al. (2021); Chen et al. (2022a); Zhao et al. (2023); Ma et al. (2023) integrate several SE methods with GT to incorporate critical structural information such as node centrality, shortest path distance, and graph substructures. Moreover, Kreuzer et al. (2021); Dwivedi et al. (2022); Rampasek et al. (2022); Bo et al. (2023) introduce various PE approaches based upon Laplacian eigenpairs. Nonetheless, these methods generally involve nontrivial overheads to compute PE/SE, and mostly adopt the self-attention module in the vanilla Transformer model that has quadratic complexity with respect to the number of nodes in a graph, prohibiting their applications in large-scale node classification tasks. To address the scalability challenge, many linear GTs have been recently proposed. Concretely, Choromanski et al. (2021); Zhang et al. (2022); Shirzad et al. (2023); Kong et al. (2023) aim to sparsify the self-attention matrix via leveraging node sampling or expander graphs, while Wu et al. (2022; 2023) focus on kernel-based approximations on the self-attention matrix. Unfor-

tunately, both prior work (Platonov et al., 2023) and our empirical results indicate that those linear GT models still underperform state-of-the-art GNN counterparts on several popular datasets, which poses a serious concern regarding the practical advantages of linear GTs over GNNs.

In this work, we provide an orthogonal way to ease the tension between expressivity and scalability of GTs. Specifically, we propose Polynormer, a linear GT model that is polynomial-expressive: an $L$-layer Polynormer can expressively represent a polynomial of degree $2^L$, which maps input node features to output node representations and is equivariant to node permutations. Note that the polynomial expressivity is well motivated by the Weierstrass theorem which guarantees any smooth function can be approximated by a polynomial (Stone, 1948). To this end, we first introduce a base attention model that explicitly learns a polynomial function whose coefficients are determined by the attention scores among nodes. By imposing the permutation equivariance constraint on polynomial coefficients based on graph topology and node features separately, we derive local and global attention models respectively from the base model. Subsequently, Polynormer adopts a linear local-to-global attention paradigm for learning node representations, which is a common practice for efficient transformers in language and vision domains yet less explored on graphs. To demonstrate the efficacy of Polynormer, we conduct extensive experiments by comparing Polynormer against 22 competitive GNNs and GTs on 13 node classification datasets that include homophilic and heterophilic graphs with up to millions of nodes. We believe this is possibly one of the most extensive comparisons in literature. Our main technical contributions are summarized as follows:

• To the best of our knowledge, we are the first to propose a polynomial-expressive graph transformer, which is achieved by introducing a novel attention model that explicitly learns a high-degree polynomial function with its coefficients controlled by attention scores.

• By integrating graph topology and node features into polynomial coefficients separately, we derive local and global equivariant attention modules. As a result, Polynormer harnesses the local-to-global attention mechanism to learn polynomials that are equivariant to node permutations.

• Owing to the high polynomial expressivity, Polynormer without any activation function is able to surpass state-of-the-art GNN and GT baselines on multiple datasets. When further combined with $ReLU$ activation, Polynormer improves accuracy over those baselines by a margin of up to $4.06\%$ across 11 out of 13 node classification datasets, including both homophilic and heterophilic graphs.

• Through circumventing the computation of dense attention matrices and the expensive PE/SE methods used by prior arts, our local-to-global attention scheme has linear complexity in regard to the graph size. This renders Polynormer scalable to large graphs with millions of nodes.

## 2 BACKGROUND

There is an active body of research on GTs and polynomial networks, from which we draw inspiration to build a polynomial-expressive GT with linear complexity. In the following, we present preliminaries for both areas and provide an overview of their related work.

**Graph transformers** exploit the Transformer architecture (Vaswani et al., 2017) on graphs. Specifically, given an $n$-node graph $\mathcal{G}$ and its node feature matrix $\boldsymbol{X} \in \mathbb{R}^{n \times d}$, where $d$ represents the node feature dimension, a GT layer first projects $\boldsymbol{X}$ into query, key, and value matrices, i.e., $\boldsymbol{Q} = \boldsymbol{X}\boldsymbol{W}_Q, \boldsymbol{K} = \boldsymbol{X}\boldsymbol{W}_K, \boldsymbol{V} = \boldsymbol{X}\boldsymbol{W}_V$, where $\boldsymbol{W}_Q, \boldsymbol{W}_K, \boldsymbol{W}_V \in \mathbb{R}^{d \times d}$ are three trainable weight matrices. Subsequently, the output $\boldsymbol{X}'$ with self-attention is calculated as:

$$\boldsymbol{S} = \frac{\boldsymbol{Q}\boldsymbol{K}^T}{\sqrt{d}}, \boldsymbol{X}' = softmax(\boldsymbol{S})\boldsymbol{V} \tag{1}$$

where $\boldsymbol{S} \in \mathbb{R}^{n \times n}$ is the self-attention matrix. For simplicity of illustration, we assume query, key, and value have the same dimension and only consider the single-head self-attention without bias terms. The extension to the multi-head attention is standard and straightforward.

As Equation 1 completely ignores the graph topology, various PE/SE methods have been proposed to incorporate the critical graph structural information into GTs. Concretely, Kreuzer et al. (2021); Dwivedi et al. (2022); Rampasek et al. (2022); Bo et al. (2023) use the top-$k$ Laplacian eigenpairs as node PEs, while requiring nontrivial computational costs to learn the sign ambiguity of Laplacian eigenvectors. Similarly, SE methods also suffer from high complexity for computing the distance of all node pairs or sampling graph substructures (Ying et al., 2021; Chen et al., 2022a; Zhao et al.,

2023; Ma et al., 2023). Apart from the expensive PE/SE computation, most of these approaches follow Equation 1 to compute the dense attention matrix $\boldsymbol{S}$, resulting their quadratic complexity with regard to the number of nodes. While there are scalable GTs recently proposed by linearizing the attention matrix and not involving PE/SE, they lack a thorough analysis of their expressivity in practice and may perform worse than state-of-the-art GNNs (Choromanski et al., 2021; Zhang et al., 2022; Shirzad et al., 2023; Kong et al., 2023; Wu et al., 2022; 2023). In this work, we provide a novel way to balance the expressivity and scalability of GTs, via introducing a linear GT model that can expressively represent a high-degree polynomial.

**Polynomial networks** aim to learn a function approximator where each element of the output is expressed as a polynomial of the input features. Formally, we denote the mode-$m$ vector product of a tensor $\mathsf{T} \in \mathbb{R}^{I_1 \times I_2 \times \cdots \times I_M}$ with a vector $\boldsymbol{u} \in \mathbb{R}^{I_m}$ by $\mathsf{T} \times_m \boldsymbol{u}$. Given the input feature $\boldsymbol{x} \in \mathbb{R}^n$ and the output $\boldsymbol{y} \in \mathbb{R}^o$, polynomial networks learn a polynomial $\mathcal{P} : \mathbb{R}^n \to \mathbb{R}^o$ with degree $R \in \mathbb{N}$:

$$\boldsymbol{y}_j = \mathcal{P}(\boldsymbol{x})_j = \boldsymbol{b}_j + {\boldsymbol{w}_j^{[1]}}^T \boldsymbol{x} + \boldsymbol{x}^T \boldsymbol{W}_j^{[2]} \boldsymbol{x} + \mathsf{W}_j^{[3]} \times_1 \boldsymbol{x} \times_2 \boldsymbol{x} \times_3 \boldsymbol{x} + \cdots + \mathsf{W}_j^{[R]} \prod_{r=1}^{R} \times_r \boldsymbol{x} \quad (2)$$

where $\boldsymbol{b}_j \in \mathbb{R}$ and $\{\mathsf{W}_j^{[r]} \in \mathbb{R}^{\overbrace{n \times n \times \cdots \times n}^{r \text{ times}}}\}_{r=1}^R$ are learnable parameters for the $j$-th element of output $\boldsymbol{y}$. Note that Equation 2 can be naturally extended to a more general case where both the input and output of the polynomial $\mathcal{P}$ are matrices or higher-order tensors. Moreover, suppose $\boldsymbol{x}$ and $\boldsymbol{y}$ have the same dimension (e.g., $\boldsymbol{x}, \boldsymbol{y} \in \mathbb{R}^n$), we say a polynomial $\mathcal{P}$ is *permutation equivariant* if for any permutation $\alpha$ of the indices $\{1, 2, \cdots, n\}$, the following property holds:

$$\mathcal{P}(\alpha \cdot \boldsymbol{x}) = \alpha \cdot \mathcal{P}(\boldsymbol{x}) = \alpha \cdot \boldsymbol{y} \quad (3)$$

Note that when $\boldsymbol{x} \in \mathbb{R}^n$ represents node features for an $n$-node graph (i.e., each node feature has a scalar value), the aforementioned equivariance property essentially indicates the polynomial $\mathcal{P}$ is equivariant to node permutations. It is worth mentioning that the polynomial networks are fundamentally distinct from the polynomial graph filtering methods widely explored in spectral-based GNNs, which purely focus on the polynomials of graph shift operators rather than node features.

In the literature, the notion of learnable polynomial functions can be traced back to the group method of data handling (GMDH), which learns partial descriptors that capture quadratic correlations between specific pairs of input features (Ivakhnenko, 1971). Later, the pi-sigma network (Shin & Ghosh, 1991) and its extensions (Voutriaridis et al., 2003; Li, 2003) have been proposed to capture higher-order feature interactions, while they struggle with scaling to high-dimensional input features. Recently, Chrysos et al. (2020) introduce P-nets that utilize a special kind of skip connections to efficiently implement the polynomial expansion with high-dimensional features. In addition, Chrysos et al. (2022) express as polynomials a collection of popular neural networks, such as AlexNet (Krizhevsky et al., 2012), ResNet (He et al., 2016), and SENet (Hu et al., 2018), and improve their polynomial expressivity accordingly. While these approaches have demonstrated promising results on image and audio classification tasks, they are not directly applicable to graphs.

To learn polynomials on graph-structured data, Maron et al. (2018) first introduce the basis of constant and linear functions on graphs that are equivariant to node permutations. Subsequently, a series of expressive graph models have been developed by leveraging polynomial functions (Maron et al., 2019; Chen et al., 2019; Azizian & Lelarge, 2020; Hua et al., 2022). More recently, Puny et al. (2023) demonstrate that the polynomial expressivity is a finer-grained measure than the traditional Weisfeiler-Lehman (WL) hierarchy for assessing the expressive power of graph learning models. Besides, they devise a graph polynomial model that achieves strictly better than 3-WL expressive power with quadratic complexity. However, these prior studies either only consider polynomials on local structures or cannot scale to large graphs due to their high complexity. In contrast, our work learns a high-degree polynomial on node features while integrating graph topology into polynomial coefficients. As a result, the learned polynomial function captures both local and global structural information with linear complexity, rendering it applicable to large-scale graphs.

## 3 METHODOLOGY

In this work, we follow the common setting that there are a graph $\mathcal{G} = (\mathcal{V}, \mathcal{E})$ and its node feature matrix $\boldsymbol{X} \in \mathbb{R}^{n \times d}$, where $n$ and $d$ denote the number of nodes and node feature dimension, respectively. Our goal is to design a polynomial-expressive GT model $\mathcal{F}$ that produces node representations $\boldsymbol{Y} = \mathcal{F}(\mathcal{G}, \boldsymbol{X}) = \mathcal{P}_{\mathcal{G}}(\boldsymbol{X})$, where $\mathcal{P}_{\mathcal{G}}$ is a high-degree polynomial on $\boldsymbol{X}$ whose learnable

$$y = (Wx) \odot (x + b)$$

$y_1 = W_{1,1}b_1x_1 + W_{1,2}b_1x_2 + W_{1,3}b_1x_3 + W_{1,1}x_1x_1 + W_{1,2}x_1x_2 + W_{1,3}x_1x_3$

$y_2 = W_{2,1}b_2x_1 + W_{2,2}b_2x_2 + W_{2,3}b_2x_3 + W_{2,1}x_2x_1 + W_{2,2}x_2x_2 + W_{2,3}x_2x_3$

$y_3 = W_{3,1}b_3x_1 + W_{3,2}b_3x_2 + W_{3,3}b_3x_3 + W_{3,1}x_3x_1 + W_{3,2}x_3x_2 + W_{3,3}x_3x_3$

Figure 1: A toy example on a 3-node graph with scalar node features.

coefficients encode the information of $\mathcal{G}$. To this end, we first introduce a base attention model in Section 3.1 that explicitly learns high-degree polynomials. To enable the learned polynomial equivariant to node permutations, we extend the base model to equivariant local and global (linear) attention models in Section 3.2, via incorporating graph topology and node features respectively. Finally, Section 3.3 presents the proposed Polynormer architecture that employs a local-to-global attention scheme based on the equivariant attention models. As a result, Polynormer preserves high polynomial expressivity while simultaneously benefiting from linear complexity.

## 3.1 A POLYNOMIAL-EXPRESSIVE BASE MODEL WITH ATTENTION

By following the concept of polynomial networks in Hua et al. (2022); Chrysos et al. (2020; 2022), we provide Definition 3.1 of the polynomial expressivity, which measures the capability of learning high-degree polynomial functions that map input node features into output node representations. Appendix M provides a detailed discussion on our definition and comparison to prior work.

**Definition 3.1.** Given an $n$-node graph with node features $\boldsymbol{X} \in \mathbb{R}^{n \times d}$, a model $\mathcal{P} : \mathbb{R}^{n \times d} \to \mathbb{R}^{n \times d}$ is *r-polynomial-expressive* if for any node $i$ and degree-$(r-1)$ monomial $M^{r-1}$ formed by rows in $\boldsymbol{X}$, $\mathcal{P}$ can be parameterized such that $\mathcal{P}(\boldsymbol{X})_i = \boldsymbol{X}_i \odot M^{r-1}$, where $\odot$ denotes the Hadamard product.

**Polynomial expressivity of prior graph models.** For typical message-passing GNN models, we can unify their convolution layer as: $\mathcal{P}(\boldsymbol{X})_i = \sum_j c_{i,j} \boldsymbol{X}_j$, where $c_{i,j}$ is the edge weight between nodes $i$ and $j$. As each output is a linear combination of input node features, these models are at most 1-polynomial-expressive. Hence, they mainly rely on the activation function to capture nonlinearity, instead of explicitly learning high-degree polynomials. In regard to GTs and high-order GNNs (e.g., gating-based GNNs), they only capture a subset of all possible monomials with a certain degree, resulting in their limited polynomial expressivity, as discussed in Appendix C

**A motivating example.** Before constructing a polynomial-expressive model, let us first examine a simplified scenario where node features $\boldsymbol{x} \in \mathbb{R}^n$, i.e., each node feature is a scalar. In this context, we consider a model $\mathcal{P}$ that outputs $\boldsymbol{y} = \mathcal{P}(\boldsymbol{x}) = (\boldsymbol{W}\boldsymbol{x}) \odot (\boldsymbol{x} + \boldsymbol{b})$, where $\boldsymbol{W} \in \mathbb{R}^{n \times n}$ and $\boldsymbol{b} \in \mathbb{R}^n$ are weight matrices. Figure 1 shows that this model $\mathcal{P}$ is able to represent degree-2 polynomials, which consist of all possible monomials of degree up to 2 (except the constant term). Besides, as $\boldsymbol{W}$ controls the coefficients of quadratic monomials (i.e., $\boldsymbol{x}_1\boldsymbol{x}_2$, $\boldsymbol{x}_1\boldsymbol{x}_3$, and $\boldsymbol{x}_2\boldsymbol{x}_3$), we can interpret $\boldsymbol{W}$ as a general attention matrix, where $\boldsymbol{W}_{i,j}$ represents the importance of node $j$ to node $i$.

Motivated by the above example, we proceed to establish a base model using the following definition; we then theoretically analyze its polynomial expressivity.

**Definition 3.2.** Given the input node features $\boldsymbol{X}^{(0)} \in \mathbb{R}^{n \times d}$ and trainable weight matrices $\boldsymbol{W} \in \mathbb{R}^{n \times n}, \boldsymbol{B} \in \mathbb{R}^{n \times d}$, a model $\mathcal{P}$ is defined as the *base model* if its $l$-th layer is computed as:

$$\boldsymbol{X}^{(l)} = (\boldsymbol{W}^{(l)}\boldsymbol{X}^{(l-1)}) \odot (\boldsymbol{X}^{(l-1)} + \boldsymbol{B}^{(l)}) \tag{4}$$

**Theorem 3.3.** *An $L$-layer base model $\mathcal{P}$ is $2^L$-polynomial-expressive.*

The complete proof for Theorem 3.3 is available in Appendix A. Theorem 3.3 shows that the polynomial expressivity of the base model increases exponentially with the number of layers. Additionally, Appendices A and B reveal that $\boldsymbol{W}$ and $\boldsymbol{B}$ control the learned polynomial coefficients, which can be viewed as attention scores for measuring the importance of different node feature interactions.

**Limitations of the base model.** As the matrices $\boldsymbol{W}$ and $\boldsymbol{B}$ fail to exploit any graph inductive bias, the base model learns polynomials that are not equivariant to node permutations. Besides, the model has quadratic complexity due to the $n^2$ size of $\boldsymbol{W}$ and thus cannot scale to large graphs. To address both limitations, we derive two equivariant models with linear attention in Section 3.2.

## 3.2 Equivariant Attention Models with Polynomial Expressivity

For clarity, we omit the layer index $(l)$ in the ensuing discussion unless it is explicitly referenced. Instead of learning the non-equivariant matrix $\boldsymbol{B} \in \mathbb{R}^{n \times d}$ in Equation 4, we replace it with learnable weights $\boldsymbol{\beta} \in \mathbb{R}^d$ sharing across nodes, i.e., $\boldsymbol{B} = \mathbf{1}\boldsymbol{\beta}^{\boldsymbol{T}}$, where $\mathbf{1} \in \mathbb{R}^n$ denotes the all-ones vector. In addition, we apply linear projections on $\boldsymbol{X}$ to allow interactions among feature channels. This leads to Equation 5, where $\boldsymbol{V} = \boldsymbol{X}\boldsymbol{W}_V$ and $\boldsymbol{H} = \boldsymbol{X}\boldsymbol{W}_H$ with trainable matrices $\boldsymbol{W}_V, \boldsymbol{W}_H \in \mathbb{R}^{d \times d}$.

$$\boldsymbol{X} = (\boldsymbol{W}\boldsymbol{V}) \odot (\boldsymbol{H} + \mathbf{1}\boldsymbol{\beta}^{\boldsymbol{T}}) \tag{5}$$

Subsequently, it is straightforward to show that we only need to achieve permutation equivariance on the term $\boldsymbol{W}\boldsymbol{V}$ in Equation 5 to build an equivariant model. To this end, we introduce the following two equivariant attention models, by leveraging graph topology and node features respectively.

**Equivariant local attention.** We incorporate graph topology information by setting $\boldsymbol{W} = \boldsymbol{A}$, where $\boldsymbol{A}$ is a (general) sparse attention matrix such that the nonzero elements in $\boldsymbol{A}$ represent attention scores of adjacent nodes. While $\boldsymbol{A}$ can be implemented by adopting any sparse attention methods previously proposed, we choose the GAT attention scheme (Veličković et al., 2017) due to its efficient implementation. We leave the exploration of other sparse attention approaches to future work. Consequently, the term $\boldsymbol{A}\boldsymbol{V}$ holds the equivariance property, i.e., $(\boldsymbol{P}\boldsymbol{A}\boldsymbol{P}^{\boldsymbol{T}})(\boldsymbol{P}\boldsymbol{V}) = \boldsymbol{P}(\boldsymbol{A}\boldsymbol{V})$ for any node permutation matrix $\boldsymbol{P}$. It is noteworthy that replacing $\boldsymbol{W}$ with $\boldsymbol{A}$ essentially imposes a sparsity constraint on the learned polynomial of the base model, such that polynomial coefficients are nonzero if and only if the corresponding monomial terms are formed by features of nearby nodes.

**Equivariant global attention.** By setting $\boldsymbol{W} = softmax(\boldsymbol{S})$, where $\boldsymbol{S}$ denotes the global self-attention matrix defined in Equation 1, the term $\boldsymbol{W}\boldsymbol{V}$ in Equation 5 becomes $softmax(\boldsymbol{S})\boldsymbol{V}$ that is permutation equivariant (Yun et al., 2019). According to our discussion in Appendix A, an $L$-layer model based on the updated Equation 5 learns an equivariant polynomial of degree $2^L$ with the coefficients determined by attention scores in $\boldsymbol{S}$. More importantly, the learned polynomial contains all monomial basis elements that capture global and high-order feature interactions. Nonetheless, the dense attention matrix $\boldsymbol{S}$ still has quadratic complexity in regard to the number of nodes, rendering the approach not scalable to large graphs. To tackle this issue, we linearize the global attention by introducing a simple kernel trick in Equation 6, where $\boldsymbol{Q} = \boldsymbol{X}\boldsymbol{W}_Q, \boldsymbol{K} = \boldsymbol{X}\boldsymbol{W}_K$ with weight matrices $\boldsymbol{W}_Q, \boldsymbol{W}_K$, and $\sigma$ denotes the sigmoid function to guarantee the attention scores are positive. The denominator term in Equation 6 ensures that the sum of attention scores is normalized to 1 per node. In this way, we preserve two key properties of the $softmax$ function: non-negativity and normalization. As we can first compute $\sigma(\boldsymbol{K}^T)\boldsymbol{V}$ whose output is then multiplied by $\sigma(\boldsymbol{Q})$, Equation 6 avoids computing the dense attention matrix, resulting in the linear global attention. In Appendix D, we further demonstrate the advantages of our approach over prior kernel-based linear attention methods in terms of hyperparameter tuning and training stability on large graphs.

$$\boldsymbol{W}\boldsymbol{V} = \frac{\sigma(\boldsymbol{Q})\sigma(\boldsymbol{K}^T)}{\sigma(\boldsymbol{Q})\sum_i \sigma(\boldsymbol{K}_{i,:}^T)}\boldsymbol{V} = \frac{\sigma(\boldsymbol{Q})(\sigma(\boldsymbol{K}^T)\boldsymbol{V})}{\sigma(\boldsymbol{Q})\sum_i \sigma(\boldsymbol{K}_{i,:}^T)} \tag{6}$$

**Complexity analysis.** Given a graph with $n$ nodes and $m$ edges, suppose the hidden dimension is $d \ll n$, then the local attention model has the complexity of $O(md + nd^2)$. Since we exploit the kernel trick to linearize the computation of global attention in Equation 6, the complexity of global attention model is reduced from $O(n^2 d)$ to $O(nd^2)$. Hence, both proposed equivariant attention models have linear complexity with respect to the number of nodes/edges.

**Discussion.** It is noteworthy that the introduced local and global attention models are intended to enable the base model to learn high-degree equivariant polynomials, which is fundamentally distinct from the purpose of prior attention models in literature. This allows our proposed Polynormer model to outperform those attention-based baselines, even without nonlinear activation functions.

## 3.3 The Polynormer Architecture

Guided by the aforementioned equivariant models with linear attention, Figure 2(b) shows the Polynormer architecture that adopts a local-to-global attention scheme based on the following modules:

**Local attention module.** We employ the local attention approach via replacing $\boldsymbol{W}$ in Equation 5 with the sparse attention matrix $\boldsymbol{A}$ as discussed in Section 3.2, which explicitly learns an equivariant

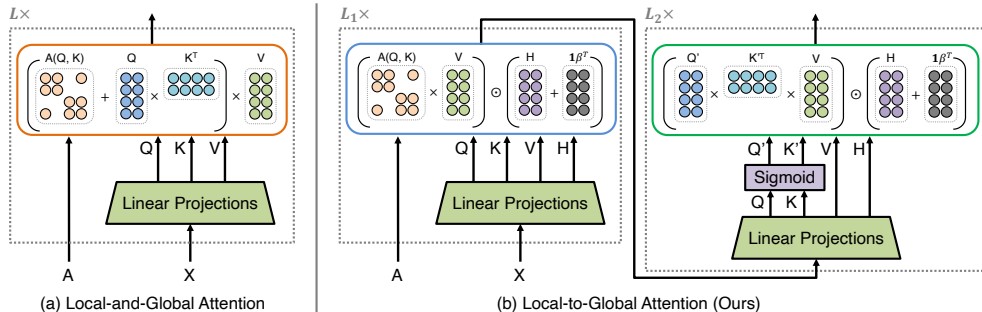

(a) Local-and-Global Attention

(b) Local-to-Global Attention (Ours)

Figure 2: Distinctions between the attention schemes in previous work and Polynormer — (a) Prior GTs use a local-and-global attention scheme, involving the simultaneous use of local and global attentions in each layer; (b) We adopt a local-to-global attention scheme, where local and global attention modules are applied sequentially; Note that the $softmax$ operator in (a) and the attention normalization in (b) are omitted for brevity.

polynomial with coefficients controlled by $A$ and $\beta$. Since the attention scores in $A$ are normalized in the range of $(0, 1)$, we apply a sigmoid function $\sigma$ on $\beta$ to scale it into the same range as $A$, which prevents $\beta$ from dominating the polynomial coefficients. This results in the local attention layer of Polynormer shown in Equation 7. By stacking $L_1$ layers, the local attention module outputs $X_{local} = \sum_{l=1}^{L_1} X^{(l)}$, which captures equivariant polynomials with different degrees based on local attention. $X_{local}$ then serves as the input for the following global attention module.

$$X = AV \odot (H + \sigma(\mathbf{1}\beta^T)) \tag{7}$$

**Global attention module.** By substituting Equation 6 into Equation 5, we obtain the global attention layer shown in Equation 8. After stacking $L_2$ global attention layers, the global attention module takes as input $X_{local}$ and produces the final node representations for downstream tasks.

$$X = \frac{\sigma(Q)(\sigma(K^T)V)}{\sigma(Q)\sum_i \sigma(K_{i,:}^T)} \odot (H + \sigma(\mathbf{1}\beta^T)) \tag{8}$$

Note that the Polynormer architecture described above does not apply any activation function on the outputs per layer. Inspired by prior polynomial networks on images (Chrysos et al., 2022), we can optionally integrate $ReLU$ activation into each layer, which potentially further improves the model performance by involving additional nonlinearity. We denote this version by Polynormer-r.

**Discussion.** Compared to the popular architecture of prior GTs shown in Figure 2(a), Polynormer builds upon our proposed base model, thus preserving high polynomial expressivity. Besides, our local-to-global attention scheme outperforms the prior local-and-global attention scheme, as empirically demonstrated in Section 4.3. Another important advantage of Polynormer is its ease of implementation. By avoiding the enumeration of equivariant polynomials, Polynormer offers a concise way to learn high-degree equivariant polynomials, which can be easily implemented using popular graph learning frameworks such as PyG (Fey & Lenssen, 2019) and DGL (Wang et al., 2019). We further provide an analysis of Polynormer expressivity under the WL hierarchy in Appendix K.

## 4 EXPERIMENTS

We have conducted an extensive evaluation of Polynormer against state-of-the-art (SOTA) GNN and GT models on both homophilic and heterophilic graphs, where nearby nodes tend to have the same or different labels. Besides, we demonstrate the scalability of Polynormer on large-scale graphs that contain millions of nodes. In addition, we perform additional ablation analysis to understand the effectiveness of the local-to-global attention scheme adopted by Polynormer. Finally, we visualize Polynormer attention scores to showcase its ability of capturing critical global structures.

**Experimental Setup.** Appendix E shows the details of all 13 datasets used in our experiments, which consist of 7 homophilic and 6 heterophilic graphs. Notably, the heterophilic datasets are from Platonov et al. (2023); Lim et al. (2021), which have addressed serious issues (e.g., train-test data leakage) of conventional popular datasets such as chameleon and squirrel. We compare Polynormer against 10 competitive GNNs. Note that prior graph polynomial models except tGNN (Hua et al., 2022) discussed in Section 2 run out of GPU memory even on the smallest dataset in our

experiments. Besides, we also evaluate 6 GTs that have shown promising results on the node classification task. We report the performance results of baselines from their original papers or official leaderboards whenever possible, as those results are obtained by well-tuned models. For baselines whose results are not publicly available on given datasets, we tune their hyperparameters to achieve the highest possible accuracy. Detailed hyperparameter settings of baselines and Polynormer are available in Appendix H. Our hardware information is provided in Appendix F.

## 4.1 PERFORMANCE ON HOMOPHILIC AND HETEROPHILIC GRAPHS

Table 1: Averaged node classification accuracy (%) $\pm$ std over 10 runs on homophilic datasets. — Polynormer-r denotes Polynormer with $ReLU$ activation. We highlight the top **first**, **second**, and **third** results per dataset.

|  | Computer | Photo | CS | Physics | WikiCS |
|---|---|---|---|---|---|
| GCN | $89.65 \pm 0.52$ | $92.70 \pm 0.20$ | $92.92 \pm 0.12$ | $96.18 \pm 0.07$ | $77.47 \pm 0.85$ |
| GraphSAGE | $91.20 \pm 0.29$ | $94.59 \pm 0.14$ | $93.91 \pm 0.13$ | $96.49 \pm 0.06$ | $74.77 \pm 0.95$ |
| GAT | $90.78 \pm 0.13$ | $93.87 \pm 0.11$ | $93.61 \pm 0.14$ | $96.17 \pm 0.08$ | $76.91 \pm 0.82$ |
| GCNII | $91.04 \pm 0.41$ | $94.30 \pm 0.20$ | $92.22 \pm 0.14$ | $95.97 \pm 0.11$ | $78.68 \pm 0.55$ |
| GPRGNN | $89.32 \pm 0.29$ | $94.49 \pm 0.14$ | $95.13 \pm 0.09$ | $96.85 \pm 0.08$ | $78.12 \pm 0.23$ |
| APPNP | $90.18 \pm 0.17$ | $94.32 \pm 0.14$ | $94.49 \pm 0.07$ | $96.54 \pm 0.07$ | $78.87 \pm 0.11$ |
| PPRGo | $88.69 \pm 0.21$ | $93.61 \pm 0.12$ | $92.52 \pm 0.15$ | $95.51 \pm 0.08$ | $77.89 \pm 0.42$ |
| GGCN | $91.81 \pm 0.20$ | $94.50 \pm 0.11$ | $95.25 \pm 0.05$ | $97.07 \pm 0.05$ | $78.44 \pm 0.53$ |
| OrderedGNN | $92.03 \pm 0.13$ | $95.10 \pm 0.20$ | $95.00 \pm 0.10$ | $97.00 \pm 0.08$ | $79.01 \pm 0.68$ |
| tGNN | $83.40 \pm 1.33$ | $89.92 \pm 0.72$ | $92.85 \pm 0.48$ | $96.24 \pm 0.24$ | $71.49 \pm 1.05$ |
| GraphGPS | $91.19 \pm 0.54$ | $95.06 \pm 0.13$ | $93.93 \pm 0.12$ | $97.12 \pm 0.19$ | $78.66 \pm 0.49$ |
| NAGphormer | $91.22 \pm 0.14$ | $95.49 \pm 0.11$ | $95.75 \pm 0.09$ | $97.34 \pm 0.03$ | $77.16 \pm 0.72$ |
| Exphormer | $91.47 \pm 0.17$ | $95.35 \pm 0.22$ | $94.93 \pm 0.01$ | $96.89 \pm 0.09$ | $78.54 \pm 0.49$ |
| NodeFormer | $86.98 \pm 0.62$ | $93.46 \pm 0.35$ | $95.64 \pm 0.22$ | $96.45 \pm 0.28$ | $74.73 \pm 0.94$ |
| DIFFormer | $91.99 \pm 0.76$ | $95.10 \pm 0.47$ | $94.78 \pm 0.20$ | $96.60 \pm 0.18$ | $73.46 \pm 0.56$ |
| GOAT | $90.96 \pm 0.90$ | $92.96 \pm 1.48$ | $94.21 \pm 0.38$ | $96.24 \pm 0.24$ | $77.00 \pm 0.77$ |
| Polynormer | $93.18 \pm 0.18$ | $96.11 \pm 0.23$ | $95.51 \pm 0.29$ | $97.22 \pm 0.06$ | $79.53 \pm 0.83$ |
| Polynormer-r | $93.68 \pm 0.21$ | $96.46 \pm 0.26$ | $95.53 \pm 0.16$ | $97.27 \pm 0.08$ | $80.10 \pm 0.67$ |

**Performance on homophilic graphs**. We first compare Polynormer with 16 popular baselines that include SOTA GNNs or GTs on 5 common homophilic datasets. As shown in Table 1, Polynormer is able to outperform all baselines on 3 out of 5 datasets, which clearly demonstrate the efficacy of the proposed polynomial-expressive model. Moreover, incorporating the $ReLU$ function can lead to further improvements in the accuracy of Polynormer. This is because the nonlinearity imposed by $ReLU$ introduces additional higher-order monomials, which in turn enhance the quality of node representations. It is noteworthy that the accuracy gain of Polynormer over SOTA baselines is $1.65\%$ on *Computer*, which is a nontrivial improvement given that those baselines have been finely tuned on these well-established homophilic datasets.

**Performance on heterophilic graphs**. Table 2 reports the average results over 10 runs on heterophilic graphs. Notably, the GT baselines underperform SOTA GNNs on all 5 datasets, which raises concerns about whether those prior GT models properly utilize the expressivity brought by the self-attention module. Moreover, there is no baseline model that consistently ranks among the top 3 models across 5 datasets. In contrast, by integrating the attention mechanism into the polynomial-expressive model, Polynormer surpasses all baselines on 4 out of 5 datasets. Furthermore, Polynormer-r consistently outperforms baselines across all datasets with the accuracy improvement by a margin up to $3.55\%$ (i.e., $97.46\%$ vs. $93.91\%$ on *minesweeper*).

## 4.2 PERFORMANCE ON LARGE GRAPHS

We conduct experiments on large graphs by comparing Polynormer against GTs as well as some representative GNN models. Since the graphs in *ogbn-products* and *pokec* are too large to be loaded into the GPU memory for full-batch training, we leverage the random partitioning method adopted by prior GT models Wu et al. (2022; 2023) to perform mini-batch training. As shown in Table 3, Polynormer-r outperforms all baselines on these large graphs with the accuracy gain of up to $4.06\%$. Besides, we can observe that the accuracy of Polynormer drops $1.64\%$ when removing $ReLU$ activation on *ogbn-arxiv*. Thus, we believe the nonlinearity associated with $ReLU$ is more critical on

Table 2: Averaged node classification results over 10 runs on heterophilic datasets — Accuracy is reported for roman-empire and amazon-ratings, and ROC AUC is reported for minesweeper, tolokers, and questions. Polynormer-r denotes Polynormer with $ReLU$ activation. We highlight the top **first**, **second**, and **third** results per dataset.

|  | roman-empire | amazon-ratings | minesweeper | tolokers | questions |
|---|---|---|---|---|---|
| GCN | $73.69 \pm 0.74$ | $48.70 \pm 0.63$ | $89.75 \pm 0.52$ | $83.64 \pm 0.67$ | $76.09 \pm 1.27$ |
| GraphSAGE | $85.74 \pm 0.67$ | $53.63 \pm 0.39$ | $93.51 \pm 0.57$ | $82.43 \pm 0.44$ | $76.44 \pm 0.62$ |
| GAT-sep | $88.75 \pm 0.41$ | $52.70 \pm 0.62$ | $93.91 \pm 0.35$ | $83.78 \pm 0.43$ | $76.79 \pm 0.71$ |
| H2GCN | $60.11 \pm 0.52$ | $36.47 \pm 0.23$ | $89.71 \pm 0.31$ | $73.35 \pm 1.01$ | $63.59 \pm 1.46$ |
| GPRGNN | $64.85 \pm 0.27$ | $44.88 \pm 0.34$ | $86.24 \pm 0.61$ | $72.94 \pm 0.97$ | $55.48 \pm 0.91$ |
| FSGNN | $79.92 \pm 0.56$ | $52.74 \pm 0.83$ | $90.08 \pm 0.70$ | $82.76 \pm 0.61$ | $78.86 \pm 0.92$ |
| GloGNN | $59.63 \pm 0.69$ | $36.89 \pm 0.14$ | $51.08 \pm 1.23$ | $73.39 \pm 1.17$ | $65.74 \pm 1.19$ |
| GGCN | $74.46 \pm 0.54$ | $43.00 \pm 0.32$ | $87.54 \pm 1.22$ | $77.31 \pm 1.14$ | $71.10 \pm 1.57$ |
| OrderedGNN | $77.68 \pm 0.39$ | $47.29 \pm 0.65$ | $80.58 \pm 1.08$ | $75.60 \pm 1.36$ | $75.09 \pm 1.00$ |
| $G^2$-GNN | $82.16 \pm 0.78$ | $47.93 \pm 0.58$ | $91.83 \pm 0.56$ | $82.51 \pm 0.80$ | $74.82 \pm 0.92$ |
| DIR-GNN | $91.23 \pm 0.32$ | $47.89 \pm 0.39$ | $87.05 \pm 0.69$ | $81.19 \pm 1.05$ | $76.13 \pm 1.24$ |
| tGNN | $79.95 \pm 0.75$ | $48.21 \pm 0.53$ | $91.93 \pm 0.77$ | $70.84 \pm 1.75$ | $76.38 \pm 1.79$ |
| GraphGPS | $82.00 \pm 0.61$ | $53.10 \pm 0.42$ | $90.63 \pm 0.67$ | $83.71 \pm 0.48$ | $71.73 \pm 1.47$ |
| NAGphormer | $74.34 \pm 0.77$ | $51.26 \pm 0.72$ | $84.19 \pm 0.66$ | $78.32 \pm 0.95$ | $68.17 \pm 1.53$ |
| Exphormer | $89.03 \pm 0.37$ | $53.51 \pm 0.46$ | $90.74 \pm 0.53$ | $83.77 \pm 0.78$ | $73.94 \pm 1.06$ |
| NodeFormer | $64.49 \pm 0.73$ | $43.86 \pm 0.35$ | $86.71 \pm 0.88$ | $78.10 \pm 1.03$ | $74.27 \pm 1.46$ |
| DIFFormer | $79.10 \pm 0.32$ | $47.84 \pm 0.65$ | $90.89 \pm 0.58$ | $83.57 \pm 0.68$ | $72.15 \pm 1.31$ |
| GOAT | $71.59 \pm 1.25$ | $44.61 \pm 0.50$ | $81.09 \pm 1.02$ | $83.11 \pm 1.04$ | $75.76 \pm 1.66$ |
| Polynormer | $92.13 \pm 0.50$ | $54.46 \pm 0.40$ | $96.96 \pm 0.52$ | $84.83 \pm 0.72$ | $77.95 \pm 1.06$ |
| Polynormer-r | $92.55 \pm 0.37$ | $54.81 \pm 0.49$ | $97.46 \pm 0.36$ | $85.91 \pm 0.74$ | $78.92 \pm 0.89$ |

Table 3: Averaged node classification accuracy (%) $\pm$ std on large-scale datasets — Polynormer-r denotes Polynormer with $ReLU$ activation. We highlight the top **first**, **second**, and **third** results per dataset. $OOM$ means out of memory.

|  | ogbn-arxiv | ogbn-products | pokec |
|---|---|---|---|
| GCN | $71.74 \pm 0.29$ | $75.64 \pm 0.21$ | $75.45 \pm 0.17$ |
| GAT | $72.01 \pm 0.20$ | $79.45 \pm 0.59$ | $72.23 \pm 0.18$ |
| GPRGNN | $71.10 \pm 0.12$ | $79.76 \pm 0.59$ | $78.83 \pm 0.05$ |
| LINKX | $66.18 \pm 0.33$ | $71.59 \pm 0.71$ | $82.04 \pm 0.07$ |
| GraphGPS | $70.97 \pm 0.41$ | $OOM$ | $OOM$ |
| NAGphormer | $70.13 \pm 0.55$ | $73.55 \pm 0.21$ | $76.59 \pm 0.25$ |
| Exphormer | $72.44 \pm 0.28$ | $OOM$ | $OOM$ |
| NodeFormer | $67.19 \pm 0.83$ | $72.93 \pm 0.13$ | $71.00 \pm 1.30$ |
| DIFFormer | $69.86 \pm 0.25$ | $74.16 \pm 0.31$ | $73.89 \pm 0.35$ |
| GOAT | $72.41 \pm 0.40$ | $82.00 \pm 0.43$ | $66.37 \pm 0.94$ |
| Polynormer | $71.82 \pm 0.23$ | $82.97 \pm 0.28$ | $85.95 \pm 0.07$ |
| Polynormer-r | $73.46 \pm 0.16$ | $83.82 \pm 0.11$ | $86.10 \pm 0.05$ |

*ogbn-arxiv*, which is known to be a challenging dataset (Shirzad et al., 2023). In addition, Appendix J provides the training time and memory usage of Polynormer and GT baselines. Moreover, we further evaluate Polynormer on an industrial-level graph benchmark named TpuGraphs (Phothilimthana et al., 2024), whose results are provided in Appendix L.

## 4.3 ABLATION ANALYSIS ON POLYNORMER ATTENTION SCHEMES

Figure 3 shows the comparison of the SOTA baseline (red bar), Polynormer without global attention (orange bar), a variant of Polynormer where the local and global attention modules are applied simultaneously to update node features per layer (blue bar), and Polynormer (green bar). We provide the detailed architecture of the Polynormer variant in Appendix G. Besides, we omit the results of Polynormer without local attention, as it completely ignores the graph structural information and thus performs poorly on the graph datasets.

By comparing the orange and green bars in Figure 3, we can observe that the local attention model achieves comparable results to Polynormer on homophilic graphs (*Computer* and *Photo*), while it lags behind Polynormer on heterophilic graphs (*roman-empire* and *minesweeper*). This observation

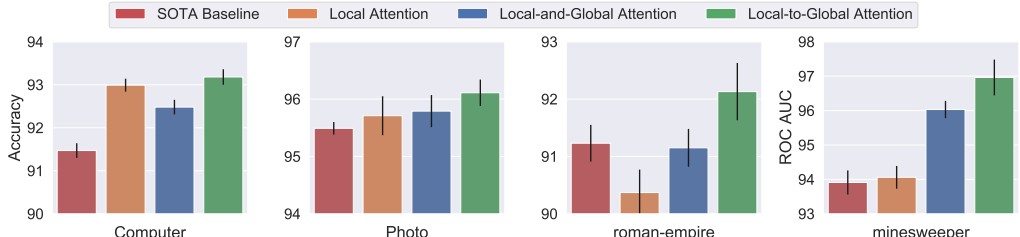

Figure 3: Ablation studies on attention modules of Polynormer — "Local Attention" means only the local attention module is used. "Local-and-Global Attention" denotes the local and global attention modules are employed in parallel, while "Local-to-Global Attention" represents our proposed Polynormer model where the local attention module is followed by the global attention module.

highlights the efficacy of the global attention module in Polynormer, which captures global information that proves more beneficial for heterophilic graphs (Li et al., 2022), as further illustrated in Section 4.4. Moreover, Figure 3 also demonstrates that the local-to-global attention adopted by Polynormer performs better than its counterpart, i.e., local-and-global attention, which has been widely used by prior GT models (Rampasek et al., 2022; Wu et al., 2022; 2023; Kong et al., 2023). We attribute it to that the local-and-global attention may introduce the risk of mixing local and non-local interactions (Di Giovanni et al., 2023), while the local-to-global attention inherently avoids this issue. This is further confirmed by the results on *Computer*, where the local-and-global attention model performs even worse than the local attention alone model.

## 4.4 VISUALIZATION



Figure 4: Visualization on the importance of nodes (columns) to each target node (row) — Higher heatmap values indicate greater importance; Both subfigures (a) and (b) consider nodes are important if they share the same label as the target node, while (a) has an additional constraint that these nodes are at most 5-hop away from the target node; Subfigure (c) measures node importance based on the corresponding global attention scores in Polynormer.

For the sake of clear visualization, we randomly sample 100 nodes from *minesweeper*, based on which three heatmaps are drawn in Figure 4. Notably, the coordinates in each heatmap correspond to node indices in the graph. We first draw Figure 4(a) by assigning every heatmap value to 1 if the corresponding two nodes have the same label and their shortest path distance is up to 5, and 0 otherwise. This essentially highlights locally important neighbors per node. Similarly, we further visualize globally important nodes by removing the local constraint, resulting in Figure 4(b). Figure 4(c) visualizes the attention matrix in the last global attention layer of Polynormer. Note that we linearly scale the attention scores such that the maximum value is 1 for the visualization purpose. By comparing Figures 4(a) and 4(b), we observe numerous globally important nodes that can be exploited for accurate predictions on target nodes. Figure 4(c) clearly showcases that the attention scores of Polynormer effectively differentiate those globally important nodes from unimportant ones. As a result, these attention scores enable Polynormer to focus on crucial monomial terms that consist of a sequence of globally important nodes, allowing it to learn critical global structural information.

## 5 CONCLUSIONS

This work introduces Polynormer that adopts a linear local-to-global attention scheme with high polynomial expressivity. Our experimental results indicate that Polynormer achieves SOTA results on a wide range of graph datasets, even without the use of nonlinear activation functions.

## ACKNOWLEDGMENTS

This work is supported in part by NSF Awards #2118709 and #2212371, a Qualcomm Innovation Fellowship, and ACE, one of the seven centers in JUMP 2.0, a Semiconductor Research Corporation (SRC) program sponsored by DARPA.

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

## A    PROOF FOR THEOREM 3.3

*Proof.* For the convenience of our proof, we denote the initial feature matrix by $\boldsymbol{X} \in \mathbb{R}^{n \times d}$, the number of layers by $L$, the number of nodes by $n$, and $[n] = \{1, 2, ..., n\}$. Besides, we set $\boldsymbol{B}^{(l)} = 0, \forall l \in [L]$. The role of $\boldsymbol{B}^{(l)}$ is analyzed in Appendix B. By defining $I$ to be an ordered set that consists of node indices (e.g., $I = (i_1, i_2, i_3, i_4)$), we define the following function $c_L$ on coefficients:

$$c_L(I; \boldsymbol{W}) = \underbrace{\boldsymbol{W}_{i_1,i_2}^{(L)}}_{2^0 \text{ times}} \underbrace{\boldsymbol{W}_{i_2,i_3}^{(L-1)} \boldsymbol{W}_{i_3,i_4}^{(L-1)}}_{2^1 \text{ times}} \cdots \underbrace{\boldsymbol{W}_{i_{2^{L-1}},i_{2^{L-1}+1}}^{(1)} \cdots \boldsymbol{W}_{i_{2^L-1},i_{2^L}}^{(1)}}_{2^{L-1} \text{ times}} \tag{9}$$

where $\boldsymbol{W}_{i_a,i_b}^{(j)}$ represents the $(i_a, i_b)$-th block in $\boldsymbol{W}$ at the $j$-th layer. Based on Equation 9, we further define the following monomial $f_L$ of degree $2^L$, where each $\boldsymbol{X}_i$ denotes a $d$-dimensional feature vector of node $i$:

$$f_L(I; \boldsymbol{W}, \boldsymbol{X}) = c_L(I; \boldsymbol{W}) \, \boldsymbol{X}_{i_1} \odot \boldsymbol{X}_{i_2} \cdots \odot \boldsymbol{X}_{i_{2^L}} \tag{10}$$

Next, we prove by induction that the $L$-layer base model produces the following node representation $\boldsymbol{X}_i^{(L)}$ for any node $i \in [n]$:

$$\boldsymbol{X}_i^{(L)} = \sum_{I \in S_{2^L}^i} f_L(I; \boldsymbol{W}, \boldsymbol{X}) \tag{11}$$

where $S_{2^L}^i$ is a set that represents all the combinations of choosing $2^L$ elements from $[n]$ (with replacement), with the first element fixed to be $i$, i.e., $S_{2^L}^i := \{y \in [n]^{2^L} \mid y_1 = i\}$.

**Base case.** When $L = 1$, we have:

$$\begin{aligned}
\boldsymbol{X}_i^{(1)} &= (\sum_{j=1}^n \boldsymbol{W}_{i,j}^{(1)} \boldsymbol{X}_j) \odot \boldsymbol{X}_i \\
&= \sum_{j=1}^n \boldsymbol{W}_{i,j}^{(1)} \, \boldsymbol{X}_i \odot \boldsymbol{X}_j \\
&= \sum_{I \in S_{2^1}^i} \boldsymbol{W}_{i_1,i_2}^{(1)} \, \boldsymbol{X}_{i_1} \odot \boldsymbol{X}_{i_2} \\
&= \sum_{I \in S_{2^1}^i} f_1(I; \boldsymbol{W}, \boldsymbol{X})
\end{aligned} \tag{12}$$

**Induction step.** When $L = l$, suppose we have $\boldsymbol{X}_i^{(l)} = \sum_{I \in S_{2^l}^i} f_l(I; \boldsymbol{W}, \boldsymbol{X})$. Then, for $L = l + 1$, we have:

$$\begin{aligned}
\boldsymbol{X}_i^{(l+1)} &= (\sum_{j=1}^n \boldsymbol{W}_{i,j}^{(l+1)} \boldsymbol{X}_j^{(l)}) \odot \boldsymbol{X}_i^{(l)} \\
&= (\sum_{j=1}^n \boldsymbol{W}_{i,j}^{(l+1)} \sum_{J \in S_{2^l}^j} f_l(J; \boldsymbol{W}, \boldsymbol{X})) \odot (\sum_{I \in S_{2^l}^i} f_l(I; \boldsymbol{W}, \boldsymbol{X})) \\
&= (\sum_{j=1}^n \boldsymbol{W}_{i,j}^{(l+1)} \sum_{J \in S_{2^l}^j} c_l(J; \boldsymbol{W}) \, \boldsymbol{X}_{j_1} \odot \boldsymbol{X}_{j_2} \cdots \odot \boldsymbol{X}_{j_{2^l}}) \\
&\quad \odot (\sum_{I \in S_{2^l}^i} c_l(I; \boldsymbol{W}) \, \boldsymbol{X}_{i_1} \odot \boldsymbol{X}_{i_2} \cdots \odot \boldsymbol{X}_{i_{2^l}}) \\
&= \sum_{I \in S_{2^l}^i} \sum_{j=1}^n \sum_{J \in S_{2^l}^j} c_l(I; \boldsymbol{W}) \, \boldsymbol{W}_{i,j}^{(l+1)} c_l(J; \boldsymbol{W}) \, \boldsymbol{X}_{i_1} \odot \boldsymbol{X}_{i_2} \cdots \odot \boldsymbol{X}_{i_{2^l}}
\end{aligned}$$

$$\odot \boldsymbol{X}_{j_1} \odot \boldsymbol{X}_{j_2} \cdots \odot \boldsymbol{X}_{j_{2^l}}$$

$$= \sum_{I \in S^i_{2^{l+1}}} c_{l+1}(I; \boldsymbol{W})\, \boldsymbol{X}_{i_1} \odot \boldsymbol{X}_{i_2} \cdots \odot \boldsymbol{X}_{i_{2^l}} \odot \boldsymbol{X}_{i_{2^l+1}} \cdots \odot \boldsymbol{X}_{i_{2^{l+1}}}$$

$$= \sum_{I \in S^i_{2^{l+1}}} f_{l+1}(I; \boldsymbol{W}, \boldsymbol{X}) \tag{13}$$

The first 3 equations and the last equation above are straightforward. For the fourth equation, we essentially expand the product in LHS to the sum in RHS. For the second last euqation, we merge the weight matrix $\boldsymbol{W}^{(l+1)}$ into the coefficient function $c_l(J; \boldsymbol{W})$, and then replace the index set $J = \{j_1, j_2, ..., j_{2^l}\}$ with $I' = \{i_{2^l+1}, i_{2^l+2}, ..., i_{2^{l+1}}\}$, which is combined with the set $\{i_1, i_2, ..., i_{2^l}\}$ to obtain the new index set $I = \{i_1, i_2, ..., i_{2^l}, i_{2^l+1}, ..., i_{2^{l+1}}\} \in S^i_{2^{l+1}}$. This complements our proof for Equation 11.

As for any node $i$, the first index of $I \in S^i_{2^L}$ is always $i$, we can rewrite Equation 11 as:

$$\boldsymbol{X}^{(L)}_i = \sum_{I \in S^i_{2^L}} f_L(I; \boldsymbol{W}, \boldsymbol{X})$$

$$= \sum_{I \in S^i_{2^L}} c_L(I; \boldsymbol{W})\, \boldsymbol{X}_{i_1} \odot \boldsymbol{X}_{i_2} \cdots \odot \boldsymbol{X}_{i_{2^L}}$$

$$= \boldsymbol{X}_{i_1} \odot \sum_{I \in S^i_{2^L}} c_L(I; \boldsymbol{W})\, \boldsymbol{X}_{i_2} \cdots \odot \boldsymbol{X}_{i_{2^L}}$$

$$= \boldsymbol{X}_i \odot \sum_{I' \in S_{2^L-1}} \boldsymbol{W}^{(L)}_{i,i'_1} \boldsymbol{W}^{(L-1)}_{i'_1,i'_2} \cdots \boldsymbol{W}^{(1)}_{i'_{2^L-2},i'_{2^L-1}}\, \boldsymbol{X}_{i'_1} \cdots \odot \boldsymbol{X}_{i'_{2^L-1}} \tag{14}$$

where $S_{2^L-1}$ is a set that represents all the combinations of choosing $2^L - 1$ elements from $[n]$ (with replacement), i.e., $S_{2^L-1} := \{y \in [n]^{2^L-1}\}$.

As a result, for any degree-($2^L$-1) monomial $M^{2^L-1}$ formed by rows in $\boldsymbol{X}$, we can denote it by $\boldsymbol{X}_{j_1} \odot \boldsymbol{X}_{j_2} \cdots \odot \boldsymbol{X}_{j_{2^L-1}}$, which corresponds to a specific oredered index set $I' \in S_{2^L-1}$ in Equation 14. Therefore, we can always parameterize weight matrices $\boldsymbol{W}^{(l)}, l \in [L]$ such that only the elements with indices $I'$ that determine the given monomial are 1 in Equation 14, and all other elements in $\boldsymbol{W}^{(l)}$ are set to 0. Consequently, Equation 14 with parameterized $\boldsymbol{W}^{(l)}$ becomes $\boldsymbol{X}_i \odot M^{2^L-1}$, which complements our proof for Theorem 3.3. $\qquad\square$

**Discussion.** Notably, Equation 9 reveals that the weight matrix $\boldsymbol{W}$ controls the coefficients of the monomial terms of degree $2^L$ in the learned polynomial. Thus, if we replace $\boldsymbol{W}$ with any attention matrix $\boldsymbol{S}$ (e.g., the global self-attention matrix described in Section 3.2), then the attention scores in $\boldsymbol{S}$ naturally control all the monomial coefficients, which essentially capture the importance of different (global) node feature interactions with order $2^L$. In Appendix B, we are going to provide the analysis to demonstrate that the matrix $\boldsymbol{B}$ in Equation 4 controls the coefficients of lower-degree monomials in practice.

## B  ANALYSIS ON $\boldsymbol{B}$ IN EQUATION 4

By ignoring the weight matrix $\mathbf{B}$ in Equation 4, we have demonstrated in Appendix A that the base model learns a polynomial that encompasses all possible monomials of degree $2^L$. However, our proof in Appendix A also reveals that the learned polynomial is unable to represent any monomials with degrees smaller than $2^L$. This raises concerns since these lower-degree monomial terms are often crucial for the model predictions (Hua et al., 2022). Fortunately, we are going to show that incorporating $\mathbf{B}$ into Equation 4 enables the $L$-layer base model to capture all monomial terms with degrees up to $2^L$ (except the constant term).

Specifically, we can consider each monomial term of the learned polynomial as generated in the following way: for each layer $l$ of the base model, we select two monomials, denoted as $M^{(l)}_1$ and

$M_2^{(l)}$, from the terms $(\boldsymbol{W}^{(l)}\boldsymbol{X}^{(l-1)})$ and $(\boldsymbol{X}^{(l-1)}+\boldsymbol{B}^{(l)})$ in Equation 4, respectively. Consequently, the generated monomial after layer $l$ is obtained by the Hadamard product, i.e., $M^{(l)} = M_1^{(l)} \odot M_2^{(l)}$. We continue this process recursively until we obtain $M^{(L)}$ at the $L$-th layer.

Regarding the selection of a monomial $M_2^{(l)}$ from $(\boldsymbol{X}^{(l-1)} + \boldsymbol{B}^{(l)})$, two scenarios arise: (i) If $M_2^{(l)}$ is a monomial term from the polynomial $\boldsymbol{X}^{(l-1)}$, it increases the degree of the generated monomial $M^{(l)}$ by up to the degree of $\boldsymbol{X}^{(l-1)}$ (i.e., $2^{l-1}$, as analyzed in Appendix A). (ii) If $M_2^{(l)} = \boldsymbol{B}^{(l)}$, it does not increase the degree.

If we never choose $M_2^{(l)} = \boldsymbol{B}^{(l)}$ for all $l \in [L]$, then the generated monomial at layer $L$ has a degree of $2^L$, as proven in Appendix A. Hence, each time we select $M_2^{(l)} = \boldsymbol{B}^{(l)}$ at layer $l$, we reduce the degree of a monomial at the $L$-th layer (whose original degree is $2^L$) by $2^{l-1}$. In other words, the more we opt for $M_2^{(l)} = \boldsymbol{B}^{(l)}$, the smaller the degree of the monomial at the $L$-th layer, and vice versa. In the extreme scenario where we choose $M_2^{(l)} = \boldsymbol{B}^{(l)}$ for all $l \in [L]$, we obtain a monomial of degree $2^L - 2^{L-1} - \cdots - 2^1 - 2^0 = 1$. Consequently, the weight matrices $\boldsymbol{B}^{(l)}, l \in [L]$, enable the base model to capture all possible monomials with degrees ranging from 1 up to $2^L$.

**Discussion.** The above analysis demonstrates the importance of $\boldsymbol{B}$ in Equation 4, i.e., controlling the coefficients of lower-degree monomials, which is another key distinction of our approach to previous gating-based GNNs that ignore $\boldsymbol{B}$ in their gating units (Yan et al., 2022; Rusch et al., 2022; Song et al., 2023).

## C  Polynomial Expressivity of Prior Graph Models

**Polynomial expressivity of GTs.** For the sake of simplicity, we omit the $softmax$ operator in GTs and assume each node feature contains a scalar value, i.e., $\boldsymbol{x} \in \mathbb{R}^n$ for an $n$-node graph. Besides, we denote the weights for query, key, and value by $w_q$, $w_k$, and $w_v$ respectively. Consequently, for any node $i$, a GT layer produces $\boldsymbol{x}_i' = w_q w_k w_v \sum_j \boldsymbol{x}_i \boldsymbol{x}_j \boldsymbol{x}_j = w_q w_k w_v \boldsymbol{x}_i \sum_j \boldsymbol{x}_j^2$, which consists of degree-3 monomials. However, there are still many degree-3 monomials that GTs fail to capture. For instance, any monomial in the set $\{\boldsymbol{x}_i^2 \boldsymbol{x}_j \mid i \neq j\}$ cannot be captured by $\boldsymbol{x}_i'$, resulting in the limited polynomial expressivity of prior GT models.

**Polynomial expressivity of high-order GNNs.** We initially focus on a closely related model known as tGNN, which is recently proposed by Hua et al. (2022). In tGNN, the polynomial for each target node $i$ is computed as $[\boldsymbol{x}_1, 1]\boldsymbol{W} \odot \cdots \odot [\boldsymbol{x}_k, 1]\boldsymbol{W}$, where $[\boldsymbol{x}_i, 1]$ represents the concatenation of the feature vector of node $i$ with a constant 1. As asserted by Hua et al. (2022), tGNN learns a multilinear polynomial in which no variables appear with a power of 2 or higher. Consequently, tGNN cannot represent any monomials where a specific variable has a power of at least 2, limiting its polynomial expressivity.

Furthermore, there exist multiple GNN models based on gating mechanisms that implicitly capture high-order interactions among node features (e.g., GGCN (Yan et al., 2022), G$^2$-GNN (Rusch et al., 2022), and OrderedGNN (Song et al., 2023)). However, these models primarily consider local structures and, therefore, are unable to learn high-order monomials formed by features of distant nodes. Additionally, few gating-based GNNs considers leveraging the weight matrix $\boldsymbol{B}$ in Equation 4 to learn lower-degree monomial terms, as discussed in Appendix B.

**Conclusion.** Due to the limitations of polynomial expressivity analyzed above, previous GTs and high-order GNNs still necessitate nonlinear activation functions to attain reasonable accuracy. In contrast, our method achieves higher accuracy than these models even without activation functions, thanks to its high polynomial expressivity.

## D  Comparison to Kernel-Based Linear Attention Models

### D.1  Hyperparameter Tuning

In literature, several linear GTs employ kernel-based methods to approximate the dense self-attention matrix within the vanilla Transformer architecture. Concretely, both GraphGPS (with

Performer-based attention) (Rampasek et al., 2022) and NodeFormer (Wu et al., 2022) utilize a kernel function based on positive random features (PRFs). However, PRFs introduce a critical hyperparameter, denoted as $m$, which determines the dimension of the transformed features. In addition to this, NodeFormer introduces another crucial hyperparameter, $\tau$, controlling the temperature in Gumbel-Softmax, which is integrated into the kernel function. Furthermore, the DiFFormer model (Wu et al., 2023) offers a choice between two types of kernel functions in practice.

Consequently, these kernel-based GTs require extensive tuning of these critical hyperparameters, which can be time-consuming, particularly when dealing with large graphs. In contrast, our proposed linear global attention, as defined in Equation 6, eliminates the need for any hyperparameters in model tuning.

### D.2 Training Stability

To the best of our knowledge, the work most similar to Equation 6 is cosFormer (Qin et al., 2022), which primarily focuses on text data rather than graphs. The key distinction between cosFormer and our proposed linear global attention lies in their choice of kernel function. CosFormer utilizes the $ReLU$ function instead of the $Sigmoid$ function $\sigma$ in Equation 6. While $ReLU$ performs well on text data containing up to $16K$ tokens per sequence, we observe that it leads to training instability when applied to large graphs with millions of nodes.

Specifically, if we substitute $ReLU$ into Equation 6, the denominator term $\sum_i \sigma(K_{i,:}^T)$ becomes $\sum_i ReLU(K_{i,:}^T)$. This modification results in a training loss of $NaN$ during our experiments on both the $ogbn\text{-}products$ and $pokec$ datasets. The issue arises because $ReLU$ only sets negative values in $\boldsymbol{K}_{i,:}$ to zero while preserving positive values. As a consequence, the term $\sum_i ReLU(K_{i,:}^T)$ accumulates potentially large positive values across millions of nodes. This leads to the denominator in Equation 6 exceeding the representational capacity of a 32-bit floating-point format and results in a $NaN$ loss during model training.

In contrast, our approach employs the $Sigmoid$ function, which maps all elements in $\boldsymbol{K}_{i,:}$ to the range $(0, 1)$. As a result, $\sum_i \sigma(K_{i,:}^T)$ does not produce excessively large values, avoiding the issue of $NaN$ loss.

### E Dataset Details

Table 4: Statistics of datasets used in our experiments.

| Dataset | Type | Homophily Score | Nodes | Edges | Classes | Features |
|---------|------|-----------------|-------|-------|---------|----------|
| Computer | Homophily | 0.700 | 13,752 | 245,861 | 10 | 767 |
| Photo | Homophily | 0.772 | 7,650 | 119,081 | 8 | 745 |
| CS | Homophily | 0.755 | 18,333 | 81,894 | 15 | 6,805 |
| Physics | Homophily | 0.847 | 34,493 | 247,962 | 5 | 8,415 |
| WikiCS | Homophily | 0.568 | 11,701 | 216,123 | 10 | 300 |
| roman-empire | Heterophily | 0.023 | 22,662 | 32,927 | 18 | 300 |
| amazon-ratings | Heterophily | 0.127 | 24,492 | 93,050 | 5 | 300 |
| minesweeper | Heterophily | 0.009 | 10,000 | 39,402 | 2 | 7 |
| tolokers | Heterophily | 0.187 | 11,758 | 519,000 | 2 | 10 |
| questions | Heterophily | 0.072 | 48,921 | 153,540 | 2 | 301 |
| ogbn-arxiv | Homophily | 0.416 | 169,343 | 1,166,243 | 40 | 128 |
| ogbn-products | Homophily | 0.459 | 2,449,029 | 61,859,140 | 47 | 100 |
| pokec | Heterophily | 0.000 | 1,632,803 | 30,622,564 | 2 | 65 |

Table 4 shows the statistics of all 13 datasets used in our experiments. The homophily score per dataset is computed based on the metric proposed by Lim et al. (2021) (higher score means more homophilic).

**Train/Valid/Test splits.** For *Computer*, *Photo*, *CS*, and *Physics* datasets, we adhere to the widely accepted practice of randomly dividing nodes into training (60%), validation (20%), and test (20%) sets (Chen et al., 2022b; Shirzad et al., 2023). As for the remaining datasets in our experiments, we

use the official splits provided in their respective papers (Mernyei & Cangea, 2020; Hu et al., 2020; Lim et al., 2021; Platonov et al., 2023).

## F  HARDWARE INFORMATION

We conduct all experiments on a Linux machine equipped with an Intel Xeon Gold 5218 CPU (featuring 8 cores @ 2.30 GHz) and 4 RTX A6000 GPUs (each with 48 GB of memory). It is worth noting that Polynormer only requires 1 GPU for training, while the remaining GPUs are used to run baseline experiments in parallel.

## G  A VARIANT OF POLYNORMER

In the following, we provide the detailed architecture of the variant of Polynormer mentioned in Section 4.3, which is built upon the local-and-global attention scheme, i.e., the local and global attention layers of Polynormer are employed in parallel to update node features.

$$X = (AV + \frac{\sigma(Q)(\sigma(K^T)V)}{\sigma(Q)\sum_i \sigma(K_{i,:}^T)}) \odot (H + \sigma(1\beta^T)) \qquad (15)$$

where matrices $Q$, $K$, $V$, $H$ are obtained by linearly projecting the node feature matrix $X$ from the previous layer. It is worth pointing out that this type of attention has been widely used by prior GT models (Rampasek et al., 2022; Wu et al., 2022; 2023; Kong et al., 2023). However, our ablation study in Section 4.3 indicates that the local-and-global attention performs worse than the local-to-global attention adopted by Polynormer.

## H  HYPERPARAMETERS SETTINGS

### H.1  BASELINE MODELS

For the homophilic datasets listed in Table 1, we present the results of GCN (Kipf & Welling, 2016), GAT (Veličković et al., 2017), APPNP (Gasteiger et al., 2018), GPRGNN (Chien et al., 2020), PPRGo (Bojchevski et al., 2020), NAGphormer (Chen et al., 2022b), and Exphormer (Shirzad et al., 2023), as reported in Chen et al. (2022b); Shirzad et al. (2023).

For the heterophilic datasets in Table 2, we provide the results of GCN, GraphSAGE (Hamilton et al., 2017), GAT-sep, H2GCN (Zhu et al., 2020), GPRGNN, FSGNN (Maurya et al., 2022), and GloGNN (Li et al., 2022), as reported in Platonov et al. (2023).

In the case of large-scale datasets listed in Table 3, we include the results of GCN, GAT, GPRGNN, LINKX (Lim et al., 2021), and GOAT (Kong et al., 2023), as reported in Hu et al. (2020); Lim et al. (2021); Zhang et al. (2022); Kong et al. (2023).

For baseline models without publicly available results on given datasets, we obtain their highest achievable accuracy through tuning critical hyperparameters as follows:

**GCNII (Chen et al., 2020).** We set the hidden dimension to $512$, the learning rate to $0.001$, and the number of epochs to $2000$. We perform hyperparameter tuning on the number of layers from $\{5, 10\}$, the dropout rate from $\{0.3, 0.5, 0.7\}$, $\alpha$ from $\{0.3, 0.5, 0.7\}$, and $\theta$ from $\{0.5, 1.0\}$.

**GGCN (Yan et al., 2022).** We set the hidden dimension to $512$, the learning rate to $0.001$, and the number of epochs to $2000$. We perform hyperparameter tuning on the number of layers from $\{5, 10\}$, the dropout rate from $\{0.3, 0.5, 0.7\}$, the decay rate $\eta$ from $\{0.5, 1.0, 1.5\}$, and the exponent from $\{2, 3\}$.

**OrderedGNN (Song et al., 2023).** We set the hidden dimension to $512$, the learning rate to $0.001$, and the number of epochs to $2000$. We perform hyperparameter tuning on the number of layers from $\{5, 10\}$, the dropout rate from $\{0.3, 0.5, 0.7\}$, and the chunk size from $\{4, 16, 64\}$.

**tGCN (Hua et al., 2022).** We set the hidden dimension to $512$, the learning rate to $0.001$, and the number of epochs to $2000$. We perform hyperparameter tuning on the number of layers from $\{2, 3\}$, the dropout rate from $\{0.3, 0.5, 0.7\}$, and the rank from $\{256, 512\}$.

**G$^2$-GNN (Rusch et al., 2022).** We set the hidden dimension to 512, the learning rate to 0.001, and the number of epochs to 2000. We perform hyperparameter tuning on the number of layers from $\{5, 10\}$, the dropout rate from $\{0.3, 0.5, 0.7\}$, and the exponent $p$ from $\{2, 3, 4\}$.

**DIR-GNN (Rossi et al., 2023).** We set the hidden dimension to 512, the learning rate to 0.001, the number of epochs to 2000, and $\alpha$ to 0.5. Besides, we choose GATConv with the type of jumping knowledge as "max". We perform hyperparameter tuning on the number of layers from $\{3, 5\}$, the dropout rate from $\{0.3, 0.5, 0.7\}$.

**GraphGPS (Rampasek et al., 2022).** We choose GAT as the MPNN layer type and Performer as the global attention layer type. We set the number of layers to 2, the number of heads to 8, the hidden dimension to 64, and the number of epochs to 2000. We perform hyperparameter tuning on the learning rate from $\{1e-4, 5e-4, 1e-3\}$, and the dropout rate from $\{0.0, 0.1, 0.2, 0.3, 0.4, 0.5\}$.

**NAGphormer (Chen et al., 2022b).** We set the hidden dimension to 512, the learning rate to 0.001, the batch size to 2000, and the number of epochs to 500. We perform hyperparameter tuning on the number of layers from $\{1, 2, 3\}$, the number of heads from $\{1, 8\}$, the number of hops from $\{3, 7, 10\}$, and the dropout rate from $\{0.0, 0.1, 0.2, 0.3, 0.4, 0.5\}$.

**Exphormer (Shirzad et al., 2023).** We choose GAT as the local model and Exphormer as the global model. We set the number of epochs to 2000 and the number of heads to 8. We perform hyperparameter tuning on the learning rate from $\{1e-4, 1e-3\}$, the number of layers from $\{2, 4\}$, the hidden dimension form $\{64, 80, 96\}$, and the dropout rate from $\{0.0, 0.1, 0.2, 0.3, 0.4, 0.5\}$.

**NodeFormer (Wu et al., 2022).** We set the number of epochs to 2000. Additionally, we perform hyperparameter tuning on the learning rate from $\{1e-4, 1e-3, 1e-2\}$, the number of layers from $\{1, 2, 3\}$, the hidden dimension from $\{32, 64, 128\}$, the number of heads from $\{1, 4\}$, M from $\{30, 50\}$, K from $\{5, 10\}$, rb_order from $\{1, 2\}$, the dropout rate from $\{0.0, 0.3\}$, and the temperature $\tau$ from $\{0.10, 0.15, 0.20, 0.25, 0.30, 0.40, 0.50\}$.

**DIFFormer (Wu et al., 2023).** We use the "simple" kernel. Moreover, we perform hyperparameter tuning on the learning rate from $\{1e-4, 1e-3, 1e-2\}$, the number of epochs from $\{500, 2000\}$, the number of layers from $\{2, 3\}$, the hidden dimension form $\{64, 128\}$, the number of heads from $\{1, 8\}$, $\alpha$ from $\{0.1, 0.2, 0.3\}$, and the dropout rate from $\{0.0, 0.1, 0.2, 0.3, 0.4, 0.5\}$.

**GOAT (Kong et al., 2023).** We set the "conv_type" to "full", the number of layers to 1 (fixed by GOAT), the number of epochs to 200, the number of centroids to 4096, the hidden dimension to 256, the dropout of feed forward layers to 0.5, and the batch size to 1024. We perform hyperparameter tuning on the learning rate from $\{1e-4, 1e-3, 1e-2\}$, the global dimension from $\{128, 256\}$, and the attention dropout rate from $\{0.0, 0.1, 0.2, 0.3, 0.4, 0.5\}$.

## H.2 POLYNORMER

Table 5: Hyperparameters of Polynormer per dataset.

| | Warm-up Epochs | Local-to-Global Epochs | Local Layers | Global Layers | Dropout |
|---|---|---|---|---|---|
| Computer | 200 | 1000 | 5 | 1 | 0.7 |
| Photo | 200 | 1000 | 7 | 2 | 0.7 |
| CS | 100 | 1500 | 5 | 2 | 0.3 |
| Physics | 100 | 1500 | 5 | 4 | 0.5 |
| WikiCS | 100 | 1000 | 7 | 2 | 0.5 |
| roman-empire | 100 | 2500 | 10 | 2 | 0.3 |
| amazon-ratings | 200 | 2500 | 10 | 1 | 0.3 |
| minesweeper | 100 | 2000 | 10 | 3 | 0.3 |
| tolokers | 100 | 800 | 7 | 2 | 0.5 |
| questions | 200 | 1500 | 5 | 3 | 0.2 |
| ogbn-arxiv | 2000 | 500 | 7 | 2 | 0.5 |
| ogbn-products | 1000 | 500 | 10 | 2 | 0.5 |
| pokec | 2000 | 500 | 7 | 2 | 0.2 |

Like the baseline models, we set the hidden dimension to $512$ and the learning rate to $0.001$. Additionally, we introduce a warm-up stage dedicated to training the local module. This step ensures that the node representations generated by the local module capture meaningful graph structural information before being passed to the global module. Moreover, we leverage $8$ heads for our attention modules. For the *ogbn-products* and *pokec* datasets, we use mini-batch training with a batch size of $100,000$ and $550,000$ respectively, while for the other datasets, we employ full batch training. Besides, we disable the local attention module on *ogbn-arxiv* and *pokec* datasets by replacing GAT (Veličković et al., 2017) with GCN (Kipf & Welling, 2016), which we empirically observe perform better with lower memory usage. In Table 5, we provide the critical hyperparameters of Polynormer used with each dataset.

**Additional implementation details**. We can rewrite Equation 7 as Equation 16:

$$\boldsymbol{X} = \boldsymbol{H} \odot \boldsymbol{AV} + \sigma(\boldsymbol{1}\boldsymbol{\beta^T}) \odot \boldsymbol{AV} \tag{16}$$

To improve training stability, we adopt a *LayerNorm* on $\boldsymbol{H} \odot \boldsymbol{AV}$. Besides, we empirically find that scaling *LayerNorm*$(\boldsymbol{H} \odot \boldsymbol{AV})$ by $1 - \sigma(\boldsymbol{1}\boldsymbol{\beta^T})$ typically improves model accuracy. Thus, we implement the local attention module as shown in Equation 17, and the global attention module follows a similar way:

$$\boldsymbol{X} = (1 - \sigma(\boldsymbol{1}\boldsymbol{\beta^T})) \odot \textit{LayerNorm}(\boldsymbol{H} \odot \boldsymbol{AV}) + \sigma(\boldsymbol{1}\boldsymbol{\beta^T}) \odot \boldsymbol{AV} \tag{17}$$

Appendix I provides the basic Polynormer implementation. We refer readers to our source code for the detailed model implementation and hyperparameter settings.

## I  MODEL IMPLEMENTATION

```
# N: the number of nodes
# M: the number of edges
# D: the node feature dimension
# L1: the number of local attention layers
# L2: the number of global attention layers

# x: input node feature matrix with shape [N, D]
# edge_index: input graph structure with shape [2, M]
# local_convs: local attention layers (e.g., GATConv from PyG)
# local_betas: trainable weights with shape [L1, D]
# global_convs: global attention layers (implemented in Code 2)
# global_betas: trainable weights with shape [L2, D]

# equivariant local attention module
x_local = 0
for i, local_conv in enumerate(local_convs):
    h = h_lins[i](x)
    beta = F.sigmoid(local_betas[i]).unsqueeze(0)
    x = local_conv(x, edge_index) * (h + beta)
    x_local += x

# equivariant global attention module
x = x_local
for i, global_conv in enumerate(global_convs):
    g = g_lins[i](x)
    beta = F.sigmoid(global_betas[i]).unsqueeze(0)
    x = global_conv(x) * (g + beta)

# output linear layer
out = pred_lin(x)

# negative log-likelihood loss calculation
y_pred = F.log_softmax(out, dim=1)
loss = criterion(y_pred[train_idx], y_true[train_idx])
```

Code 1: PyTorch-style Pseudocode for Polynormer

```
# N: the number of nodes
# M: the number of edges
# D: the node feature dimension
# H: the number of attention heads

# x: node feature matrix reshaped to [N, D]

# linear projections to get query, key, value matrices
q = q_lins[i](x) # reshaped to [N, D/H, H]
k = k_lins[i](x) # reshaped to [N, D/H, H]
v = v_lins[i](x) # reshaped to [N, D/H, H]

# map q and k to (0, 1) for kernel approximation
q = F.sigmoid(q)
k = F.sigmoid(k)

# numerator
kv = torch.einsum('nmh, ndh -> mdh', k, v)
num = torch.einsum('nmh, mdh -> ndh', q, kv) # [N, D/H, H]

# denominator
k_sum = torch.einsum('ndh -> dh', k)
den = torch.einsum('ndh, dh -> nh', q, k_sum).unsqueeze(1) # [N, 1, H]

# linear global attention based on kernel trick
x = num/den
x = x.reshape(N, D)
x = LayerNorm(x)
```

Code 2: PyTorch-style Pseudocode for Linear Global Attention Layer

## J  SCALABILITY ANALYSIS OF POLYNORMER

### J.1  PROFILING RESULTS ON SYNTHETIC GRAPHS

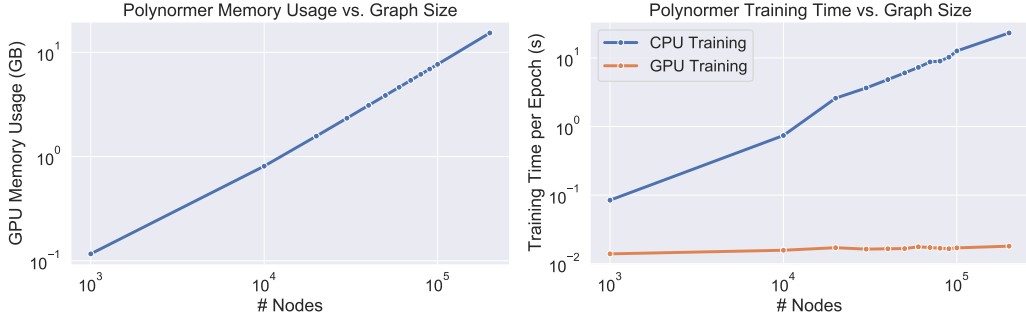

Figure 5: GPU memory usage and training time of Polynormer on synthetic graphs.

To validate the linear complexity of Polynormer with respect to the number of nodes/edges in a graph, we generate synthetic Erdos-Renyi (ER) graphs with node count ranging from $1,000$ to $200,000$. We control the edge probability in the ER model to achieve an average node degree of around $5$ and set the node feature dimension to $100$. Figure 5 (left) clearly shows that the memory usage of Polynormer linearly increases with graph size, confirming its linear space complexity. As for the time complexity of Polynormer, we observe in Figure 5 (right) that the GPU training time remains nearly constant when increasing the graph size. We attribute it to that the GPU device (RTX A6000) used in our experiments can efficiently handle parallel computations on graphs with different sizes for full batch training. This observation is further supported by the CPU training

time results, which linearly increase with graph size, as the CPU is less powerful than the GPU for parallel computation.

## J.2 PROFILING RESULTS ON LARGE REALISTIC GRAPHS

Table 6: Training time and GPU memory usage on large graphs — We underline the results obtained via full batch training, and highlight the top **first**, **second**, and **third** results per dataset.

| Method | ogbn-arxiv | | ogbn-products | | pokec | |
|---|---|---|---|---|---|---|
| | Train/Epoch (s) | Mem. (GB) | Train/Epoch (s) | Mem. (GB) | Train/Epoch (s) | Mem. (GB) |
| GAT | **0.16** | **10.64** | **0.97** | **9.04** | **1.36** | **9.57** |
| GraphGPS | 1.32 | 38.91 | *OOM* | *OOM* | *OOM* | *OOM* |
| NAGphormer | 4.26 | **5.15** | 9.64 | **7.91** | 38.32 | **6.12** |
| Exphormer | 0.74 | 34.04 | *OOM* | *OOM* | *OOM* | *OOM* |
| NodeFormer | 1.20 | 16.30 | 3.37 | 31.55 | 6.15 | 17.21 |
| DIFFormer | 0.77 | 24.51 | **1.50** | 16.24 | **3.93** | 16.03 |
| GOAT | 12.32 | **6.98** | 20.11 | **9.64** | 51.93 | **8.73** |
| Polynormer | **0.31** | 16.09 | **3.13** | 12.93 | **2.64** | 21.54 |

Table 6 shows the profiling results of GAT, linear GTs, and Polynormer in terms of training time per epoch and memory usage on large realistic graphs. Note that we perform full batch training for all models whenever possible, since it avoids the nontrivial overhead associated with graph sampling in mini-batch training.

The results demonstrate that Polynormer consistently ranks among the top 3 fastest models, with relatively low memory usage. We attribute this efficiency advantage to the implementation of Polynormer that only involves common and highly-optimized built-in APIs from modern graph learning frameworks (e.g., PyG). In contrast, the GT baselines incorporate less common or expensive compute kernels (e.g., complicated kernel functions, Gumbel-Softmax, and nearest neighbor search), making them more challenging for graph learning frameworks to accelerate. Thus, we believe Polynormer is reasonably efficient and scalable in practice.

## K  WL EXPRESSIVITY OF POLYNORMER

While we have shown that Polynormer is polynomial-expressive, it is also interesting to see its expressivity under the Weisfeiler-Lehman (WL) hierarchy. To this end, we are going to firstly show that Polynormer is at least as expressive as 1-WL GNNs, and then introduce a simple way that renders Polynormer strictly more expressive than 1-WL GNNs.

For the sake of illustration, let us revisit Equation 4 in the following:

$$X = (WX) \odot (X + B) \tag{18}$$

where $W \in \mathbb{R}^{n \times n}$, $B \in \mathbb{R}^{n \times d}$ are trainable weight matrices, and $X \in \mathbb{R}^{n \times d}$ represents node features. Suppose all nodes have identical features (followed by the original formulation of WL algorithm), if we replace $B = \mathbf{1}\beta^T$, the term $X + \mathbf{1}\beta^T$ is essentially a constant feature vector shared across nodes. As a result, the term $(WX) \odot (X + \mathbf{1}\beta^T)$ is reduced to $WX$. By properly designing the weight matrix $W$ (e.g. the adjacency matrix with self-loop), the term $WX$ can be reduced to 1-WL GNNs. Hence, Polynormer based on Equation 18 is at least as expressive as 1-WL GNNs.

To make Polynormer strictly more expressive than 1-WL GNNs, let us focus on how to properly design the weight matrix $B$. Previously, we set $B = \mathbf{1}\beta^T$. Note that the constant vector $\mathbf{1} \in \mathbb{R}^n$ here is essentially the eigenvector $v_1$ that corresponds to the smallest (normalized) graph Laplacian eigenvalue. This motivates us to design $B = v_2\beta^T$, where $v_2$ denotes the eigenvector corresponding to the second smallest (i.e., first non-trivial) eigenvalue of the normalized Laplacian. Consequently, Polynormer can be built upon the following Equation:

$$X = (WX) \odot (X + v_2\beta^T) \tag{19}$$

Notably, the vector $v_2$ essentially encodes the node positional information from graph spectrum into polynomial coefficients $B$, which makes each output node feature unique in Equation 19 and thus allows Polynormer to distinguish non-isomorphic graphs that 1-WL GNNs fail to distinguish, as empirically confirmed in Figure 6. It is noteworthy that this approach is fundamentally different from prior PE methods based on Laplacian eigenvectors. Specifically, prior PE methods primarily focus on concatenating node positional encodings with node features. In contrast, we incorporate the positional information into the polynomial coefficients learned by Polynormer.

To distinguish the aforementioned two options of designing the weight matrix $B$, we denote the Polynormer of using the constant vector $v_1$ and the second Laplacian vector $v_2$ by Polynormer-v1 and Polynormer-v2, respectively. In the following, we empirically show that Polynormer-v2 is more powerful than 1-WL GNNs on distinguishing non-isomorphic graphs.

**Experimental setup.** Without loss of generality, we focus on the local attention layer of Polynormer, and set $\beta$ as well as initial node features to be 1. As the attention scores between identical features are not meaningful, we replace the sparse attention matrix introduced in Section 3.2 with the random walk matrix $\hat{A} = AD^{-1}$, where $A$ and $D$ denote the adjacency and degree matrices, respectively.

**Experimental results.** As shown in Figure 6, Polynormer-v2 is able to distinguish non-isomorphic graphs such as circular skip link (CSL) graphs, which are known to be indistinguishable by the 1-WL test as well as 1-WL GNNs Sato (2020); Rampasek et al. (2022); Dwivedi et al. (2022).

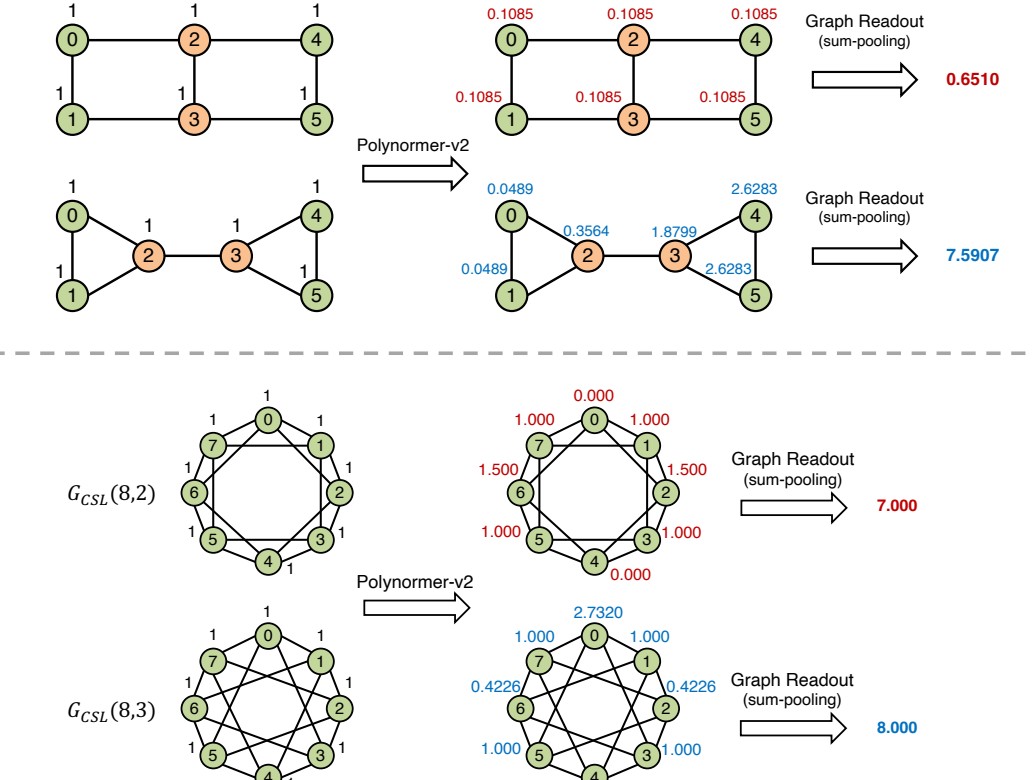

Figure 6: Two example pairs of non-isomorphic graphs from Sato (2020) that cannot be distinguished by 1-WL test. Our Polynormer-v2 can distinguish them.

While Polynormer-v2 is strictly more powerful than the 1-WL algorithm, our empirically results in Table 7 indicate that it does not clearly outperform Polynormer-v1 on realistic node classification datasets. One possible reason is that each node possesses unique features in realistic datasets, which diminishes the importance of incorporating additional node positional information into Polynormer. Hence, we implement Polynormer based on Polynormer-v1 in practice to avoid computing the Laplacian eigenvector, and leave the improvement of Polynormer-v2 to our future work.

Table 7: Comparison of Polynormer-v1 and Polynormer-v2.

|  | Computer | Photo | WikiCS | roman-empire | amazon-ratings | minesweeper |
|---|---|---|---|---|---|---|
| Polynormer-v1 | $93.18 \pm 0.18$ | $\mathbf{96.11} \pm 0.23$ | $79.53 \pm 0.83$ | $\mathbf{92.13} \pm 0.50$ | $\mathbf{54.46} \pm 0.40$ | $\mathbf{96.96} \pm 0.52$ |
| Polynormer-v2 | $\mathbf{93.25} \pm 0.22$ | $96.09 \pm 0.30$ | $\mathbf{80.01} \pm 0.55$ | $91.33 \pm 0.39$ | $54.41 \pm 0.35$ | $96.48 \pm 0.37$ |

## L  CASE STUDY: AI MODEL RUNTIME PREDICTION ON TPU

Apart from comparing Polynormer against conventional GRL models on mainstream graph datasets, we further evaluate it on TpuGraphs (Phothilimthana et al., 2024), a large-scale runtime prediction dataset on tensor computation graphs: github.com/google-research-datasets/tpu_graphs.

**Dataset details.** Unlike prior datasets for program runtime prediction that are relatively small (up to 100 nodes), TpuGraphs is a runtime prediction dataset on full tensor programs, represented as computation graphs. Each graph within the dataset represents the primary computation of an ML program, typically encompassing one or more training steps or a single inference step. These graphs are obtained from open-source ML programs and include well-known models such as ResNet, EfficientNet, Mask R-CNN, and various Transformer models designed for diverse tasks like vision, natural language processing, speech, audio, recommender systems, and generative AI. Each data sample in the dataset comprises a computational graph, a compilation configuration, and the corresponding execution time when the graph is compiled with the specified configuration on a TPU v3, an accelerator tailored for ML workloads. The compilation configuration governs how the XLA (Accelerated Linear Algebra) compiler transforms the graph through specific optimization passes. The TpuGraphs dataset is composed of two main categories based on the compiler optimization level: (1) TpuGraphs-Layout (layout optimization) and (2) TpuGraphs-Tile (tiling optimization). Layout configurations dictate how tensors are organized in physical memory, determining the dimension order for each input and output of an operation node. Notably, TpuGraphs-Layout has 4 collections based on ML model type and compiler configuration as follows:

- ML model type:
    - NLP: computation graphs of BERT models,
    - XLA: computation graphs of ML models from various domains such as vision, NLP, speech, audio, and recommendation.
- Compiler configuration:
    - Default: configurations picked by XLA compiler's heuristic.
    - Random: randomly picked configurations.

On the other hand, tile configurations control the tile size of each fused subgraph. The TpuGraphs-Layout comprises 31 million pairs of graphs and configurations, with an average of over $7,700$ nodes per graph. In contrast, the TpuGraphs-Tile includes 13 million pairs of kernels and configurations, averaging 40 nodes per kernel subgraph.

### L.1  RESULTS ON TPUGRAPHS-LAYOUT COLLECTIONS.

As shown in Table 8, we compare Polynormer against the GraphSAGE baseline provided by Google Hamilton et al. (2017). To ensure a fair comparison, we only change the model architecture while keeping all other configurations the same (i.e., no additional feature engineering). Experimental results show that Polynormer/Polynormer-local outperforms GraphSAGE on all collections. In particular, Polynormer achieves $10.3\%$ accuracy improvement over GraphSAGE and $6.3\%$ accuracy improvement over Polynormer-local on the XLA-Random collection, which is the most challenging collection since XLA consists of diverse graph structures from different domains and Random contains various compiler configurations. This showcases that the global attention of Polynormer effectively captures critical global structures in different graphs, owing to its high polynomial expressivity.

### L.2  RESULTS ON TPUGRAPHS (LAYOUT + TILE).

We further evaluate Polynormer on the whole TpuGraphs dataset that contains Layout and Tile optimizations. Table 9 shows that the local attention module alone in Polynormer is able to outperform

Table 8: Ordered pair accuracy (%) on TpuGraphs-Layout collections — "Polynormer-Local" denotes the local attention module in Polynormer.

|  | GraphSAGE | Polynormer-Local | Polynormer |
|---|---|---|---|
| NLP-Default | 81.2 | 82.0 | **82.1** |
| NLP-Random | 93.0 | **94.1** | 93.6 |
| XLA-Default | 77.3 | 73.7 | **79.6** |
| XLA-Random | 76.3 | 81.3 | **87.6** |

Table 9: Averaged runtime prediction accuracy (%) on TpuGraphs — "Polynormer-Local" denotes the local attention module in Polynormer.

|  | GraphSAGE | Polynormer-Local | Polynormer |
|---|---|---|---|
| TpuGraphs (Layout+Tile) | 30.1 | 49.8 | **61.9** |

the baseline model by 19.6%, indicating the efficacy of our polynomial-expressive architecture for capturing critical local structures. Moreover, the accuracy of Polynormer gets further improved when adding the global attention module, leading to a 31.8% accuracy improvement over the baseline model. This confirms the effectiveness of the proposed local-to-global attention scheme.

## M   DISCUSSION ON THE DEFINITION OF POLYNOMIAL EXPRESSIVITY

As graph learning models primarily learn node representations by aggregating input features from a set of nodes, they can be viewed as a multiset pooling problem with auxiliary information about the graph topology (Baek et al., 2021; Hua et al., 2022). In the context of polynomial networks on graphs, we follow a similar notion by focusing on polynomial functions that map input node features into output node representations, with the graph topology encoded into polynomial coefficients. Notably, such polynomials have been explored by prior studies Hua et al. (2022); Chrysos et al. (2020; 2022); Wang et al. (2021), which inspire us to formally provide Definition 3.1. Moreover, assessing the ability to learn high-degree polynomials is well motivated based on the Weierstrass theorem, which guarantees any smooth function can be approximated by a polynomial Stone (1948). Thus, models with higher polynomial expressivity are more capable of learning complex functions.

While Definition 3.1 does not explicitly mention graph topology, the utilization of graph topology in different graph learning models affects the inclusion of distinct monomial terms in the learned polynomial function, thereby impacting the polynomial expressivity. It is also worth noting that Definition 3.1 can be viewed as a simplified version of the definition on polynomial expressivity in Puny et al. (2023), since we consider graph structural information as polynomial coefficients rather than indeterminates. This simplification allows us to decouple the analysis on node features from graph topology, which renders us to develop a scalable model with high polynomial expressivity.

