# OpenReview forum: "Polynormer: Polynomial-Expressive Graph Transformer in Linear Time"
_ICLR.cc/2024/Conference — ICLR 2024 poster_

### Official Review · Reviewer_uwZk · 2023-10-30

**Soundness:** 2 fair
**Presentation:** 3 good
**Contribution:** 3 good
**Rating:** 6
**Confidence:** 4

**Summary:**

This paper proposes Polynormer, a polynomial- expressive GT model with linear complexity. Polynormer is built upon a novel base model that learns a high-degree polynomial on input features, with model permutation equivariance. Polynormer has been evaluated on 13 homophilic and heterophilic datasets, including large graphs with millions of nodes, with results showing that Polynormer outperforms state-of-the-art GNN and GT baselines on most datasets.

**Strengths:**

- The idea of designing a polynomial-expressive graph Transformer model is novel and interesting.

- The resulting Polynormer model is powerful, efficient, and theoretically expressive.

- The experiments are convincing, showing that Polynormer can outperform sota GNNs and GTs on a wide range of datasets.

**Weaknesses:**

- It is inappropriate to claim that GTs and GNNs has limited polynomial expressivity (in section 3.1 and appendix C), since the non-linearity layers are not negligible. In [1] it is shown that without softmax GTs cannot represent GNNs. And in [2] Transformers are proved to be universal approximators on sequences with the softmax layer as key component. Can you discuss the polynomial expressivity of GTs and GNNs with non-linearity layers? And since [2] proves that Transformers are universal approximators, do GTs have $\infty$-polynomial expressivity?

- The concept of graph is defined by edge connections. And the definition of polynomial expressivity is completely ignorant of graph structure, comparing to WL-test expressivity. From my opinion, polynomial expressivity defined here should be used to model expressivity on sets, not graphs. What is the motivation of modeling the polynomial expressivity of graph models?

- The O(N+E) complexity claim should be supported by more experiments, like a training time (VRAM) – graph size plot on synthetic random graphs with different sizes.

[1] Ying, Chengxuan, et al. "Do transformers really perform badly for graph representation?." Advances in Neural Information Processing Systems 34 (2021): 28877-28888.

[2] Yun, Chulhee, et al. "Are transformers universal approximators of sequence-to-sequence functions?." arXiv preprint arXiv:1912.10077 (2019).

**Questions:**

- See weakness above.

- Can authors discuss more about the relationship between WL-test expressivity and polynomial expressivity? For example, is the proposed Polynormer strictly more powerful than 1-WL-GNNs?

---

> ### Author Response · Authors · 2023-11-19
> **Response to Reviewer uwZk**
>
> We thank the reviewer for the constructive feedback and insightful questions on our work. Here are our detailed responses to address the concerns:
>
> **C1: Polynomial expressivity of GTs and GNNs with nonlinear functions.**
>
> A1: Firstly, we would like to note that it is important to analyze the polynomial expressivity of prior graph models without nonlinear functions, as it emphasizes the need for the activation functions in those models to achieve reasonable accuracy. This highlights Polynormer’s advantage in polynomial expressivity, which allows Polynormer to achieve promising results without the use of activation functions.
>
> While integrating nonlinear functions into prior graph models implicitly represents a high-degree polynomial, the polynomial coefficients are fixed since those nonlinear functions are typically not learnable. As a result, these models still cannot be parametrized to adaptively learn a polynomial that consists of specific monomials. In contrast, Polynormer learns the high-degree polynomial with coefficients controlled by trainable attention scores, enabling it to expressively represent various polynomial functions.
>
> **C2: Do GTs have $\infty$-polynomial expressivity?**
>
> A2: While it is true that GTs with positional encoding are universal approximators, they require unbounded depth to achieve this property. In practice, however, GTs typically have fixed and small depths (e.g., 2~3) and thus do not have $\infty$-polynomial expressivity.
>
> **C3: Why modeling polynomial expressivity of graph models?**
>
> A3: Graph models essentially learn node representations by aggregating features from a set of nodes, which can be viewed as a multiset pooling problem with auxiliary information about the graph structure [Hua-NeurIPS22; Baek-ICLR21]. Thus, the polynomial expressivity measures how well a graph model incorporates the auxiliary structural information. For instance, given a node $i$ and its $1$-hop neighbors $j$ and $k$, a model A that learns $x_i+x_j+x_k+x_ix_jx_k$ is more expressive than a model B that only learns $x_i+x_j+x_k$, since A incorporates the third-degree monomial $x_ix_jx_k$ that captures third-order structures (either a triangle or a path formed by nodes $i$, $j$, $k$).
>
> **C4: More experiments to support linear complexity.**
>
> A4: Thanks for the suggestion. In our revision, we added the training time and memory usage of Polynormer on synthetic graphs with various sizes in Appendix J.1,  which confirms that Polynormer has linear complexity with respect to the number of nodes/edges in a graph. Moreover, we further provide the complete profiling results of Polynormer and GT baselines in terms of training time and memory usage on three large graphs in Appendix J.2. The results showcase that Polynormer consistently ranks the top 3 fastest models with relatively low memory usage. This demonstrates the scalability advantage of Polynormer over prior GT models.
>
> **C5: Connection of Polynormer to WL-test expressivity.**
>
> A5: This is a great question. Indeed, it is interesting to see if Polynormer can be more powerful than 1-WL GNNs under the WL hierarchy. In response to this question, we have included Appendix K in our revision and made reference to it in Section 3.3 (marked in blue).
>
> Concretely, we show that by properly designing the weight matrix B in Equation 4, Polynormer can be more expressive than 1-WL GNNs. Initially, we set $B=\mathbf{1} \beta^T$ to build our Polynormer model. Note that the constant vector $\mathbf{1}$ here is essentially the eigenvector $v_1$ that corresponds to the smallest graph Laplacian eigenvalue. This motivates us to design $B = v_2 \beta^T$, where $v_2$ denotes the eigenvector corresponding to the second smallest (i.e., first non-trivial) Laplacian eigenvalue. To differentiate between these two choices of designing $B$, we denote the Polynormer of using $v_1$ and $v_2$ by Polynormer-v1 and Polynormer-v2, respectively. We then show that Polynormer-v2 is able to distinguish non-isomorphic graphs such as circular skip link graphs, which are known to be indistinguishable by the 1-WL test as well as 1-WL GNNs.
>
> Through empirical comparison of Polynormer-v2 with Polynormer-v1 in Table 8 (Appendix K), however, we observe similar accuracy achieved by both models on realistic node classification datasets. Hence, we implement Polynormer based on Polynormer-v1 in practice to avoid computing the Laplacian eigenvector, and leave the improvement of Polynormer-v2 to our future work.
>
> [Hua-NeurIPS22] Hua et al., “High-Order Pooling for Graph Neural Networks with Tensor Decomposition”, NeurIPS’22. \
> [Baek-ICLR21] Baek et al., “Accurate Learning of Graph Representations with Graph Multiset Pooling”, ICLR’21.

---

### Official Review · Reviewer_D5Yt · 2023-10-31

**Soundness:** 3 good
**Presentation:** 3 good
**Contribution:** 3 good
**Rating:** 6
**Confidence:** 4

**Summary:**

The proposed Polynormer is a new graph transformer model that balances expressivity and scalability. The paper introduces a polynomial model that achieves polynomial expressiveness with linear complexity, outperforming state-of-the-art GNN and GT baselines on most datasets. The model is based on a novel polynomial attention mechanism that can capture higher-order interactions between nodes in a graph. The attention mechanism is designed to be both local and global equivariant, allowing it to capture both local and global patterns in the graph.

**Strengths:**

- The idea to adopt attention model in the polynomial feature mapping is novel and interesting.
- Experiments are sufficient. Many important baselines and datasets of various sizes are covered.

**Weaknesses:**

- I find that the proposed approach (global) may also work in transformers in other fields, e.g. NLP. Could you provide such experiments to show its capacity in dealing with different types of data?
- Why and how could polynomial expressivity improve model performance? The point was not clear.

**Questions:**

- How is the degree of polynomial defined?

---

> ### Author Response · Authors · 2023-11-19
> **Response to Reviewer D5Yt**
>
> We appreciate the positive comments and interesting considerations. Here are our detailed responses:
>
> **C1: Polynormer in other fields such as NLP.**
>
> A1: We agree with the reviewer that our approach could potentially be applied in other domains (e.g., NLP), although exploring such a direction is beyond the scope of this work. We believe that introducing a novel and powerful graph transformer is a significant contribution in its own right.
>
> **C2: Why and how polynomial expressivity improves performance.**
>
> A2: High polynomial expressivity indicates that the model is capable of learning high-order interactions among node features. For instance, given a node $i$ and its $1$-hop neighbors $j$ and $k$. Conventional GNNs capture first-order (linear) feature interactions such as $x_i, x_j, x_j+x_k$ during neighbor aggregation. In contrast, a polynomial-expressive model captures additional higher-order interactions such as $x_i^2x_j, x_ix_jx_k$, etc. Notably, the high-order interactions implicitly correspond to high-order graph structures (e.g., $x_ix_jx_k$ corresponds to either a triangle or a path formed by nodes $i, j, k$.), which are known to be crucial for many graph problems.
>
> **C3: Definition of polynomial degree.**
>
> A3: The polynomial degree refers to the highest degree among monomial terms in the polynomial. In this work, the degree of a monomial term represents the number of node features for the Hadamard product, as defined in Equation 10 (Appendix A).

---

### Official Review · Reviewer_HGJh · 2023-11-02

**Soundness:** 2 fair
**Presentation:** 3 good
**Contribution:** 2 fair
**Rating:** 6
**Confidence:** 3

**Summary:**

Summary:
This paper proposes Polynormer, a graph transformer model architecture for node classification.
First, the paper introduces a base attention model in Section 3.1 that explicitly represents node features as polynomials, with coefficients determined by attention scores. This is claimed to result in high polynomial expressivity (in Sec. 3.1). To make the model equivariant, the paper integrates graph topology and node features into the polynomial coefficients to derive local and global attention models. This makes the overall Polynormer architecturem which achieves linear complexity instead of quadratic. Experiments are performed on 13 datasets including both homophilic and heterephilic tasks where Polynormer improves on most datasets.

**Strengths:**

Strengths:
- Provides theoretical analysis of polynomial expressivity, though restricted to scalar features. It goes beyond the WL expressivity as common in graph learning literature.
- Demonstrates the performance of the architecture on 13 datasets where comparisons with baselines make the proposed model better on 11 datasets.
- Ablation study on a smaller dataset group shows benefits of global attention and local-to-global scheme.

**Weaknesses:**

Weaknesses and Questions:
-The theoretical expressivity claims in Section 3.1 may be overclaiming capabilities, as proofs make simplifying assumptions about scalar features that differ from real graph data (Section 4). Can this be justified further?
-While complexity is analyzed, runtime and memory usage are not empirically compared to baselines in Section 4.2 to demonstrate scalability.

**Questions:**

included with weaknesses

---

> ### Author Response · Authors · 2023-11-19
> **Response to Reviewer HGJh**
>
> We thank the reviewer for the positive evaluation and valuable comments. Here are our detailed responses to address the concerns:
>
> **C1: Assumption on scalar node features.**
>
> A1: We would like to clarify that our proof makes no assumption about node features being scalars. Specifically, given a node feature matrix $X \in R^{n \times d}$, we first define a monomial term representing the Hadamard product of a series of $d$-dimensional node features $X_i \in R^d$, as illustrated in Equation 10 (Appendix A). Subsequently, we prove by induction in Equations 12 and 13 that Polynormer learns a polynomial which is a linear combination of the monomial terms defined in Equation 10. Thus, our proof considers high-dimensional node features.
>
> In our revision, we further highlight the shape of $X$ and $X_i$ in Appendix A (marked in blue) to indicate that our proof is not restricted to scalar node features.
>
> **C2: Runtime and memory usage on large graphs.**
>
> A2: Thanks for the suggestion. In our revision, we have added the complete profiling results of Polynormer and GT baselines in terms of training time and GPU memory usage on three large graphs in Appendix J.2, which we reference in Section 4.2 (marked in blue). The results showcase that Polynormer consistently ranks the top 3 fastest models with relatively low memory usage. This demonstrates the scalability advantage of Polynormer over prior GT models. Moreover, we also show the profiling results of Polynormer on synthetic graphs with various sizes in Appendix J.1,  which confirms that Polynormer has linear complexity with respect to the number of nodes/edges in a graph.

---

### Author Response · Authors · 2023-11-19
**Summary of Responses**

Thanks for the positive assessment and valuable comments from all reviewers! We are very encouraged that all reviewers have recognized our main contributions, including the novelty, theoretical justification, and extensive experiments. In the rebuttal, we believe we have addressed the major concerns raised by the reviewers as follows:
- We provide additional results in Appendix J to showcase the scalability advantage of Polynormer over prior graph transformers, as suggested by Reviewers HGJh and uwZk.
- We further analyze the Polynormer expressivity under the WL hierarchy in Appendix K, which demonstrates that Polynormer can be more powerful than the 1-WL test, as asked by Reviewer uwZk.
- We also clarify why high polynomial expressivity improves model performance for Reviewer D5Yt.

We have updated our manuscript accordingly to address the major concerns (marked in blue). Besides, as mentioned in our reproducibility statement, our [source code](https://drive.google.com/file/d/1ieCJLfsT9p4NPRkfteehaMXCEDdhF0k3/view?usp=sharing) is currently visible to reviewers and area chairs. The code will also become publicly available upon publication of the paper.

---

### Meta-Review · Area_Chair_6RdD · 2023-12-08

**Metareview:**

In this submission, the authors proposed a polynomial expressive graph transformer (called Polynormer), whose complexity is linear to the number of nodes and that of edges. The proposed model derives local and global attention maps jointly and learns high-order polynomial features, whose coefficients are determined by the attention maps. The authors demonstrate the equivariance of the model and further show its equivalence to the WL-1 test in the rebuttal phase. Experimental results show the effectiveness of the proposed method to some degree.

Strengths: (a) The proposed model is easy to implement, with computational efficiency. (b) The authors provide theoretical analysis, showing the permutation equivariance of the model and its relations to the WL test.

Weaknesses: (a) The presentation of the submission should be enhanced. (b) Why not take the well-known Graphormer as the baseline? (c) The reviewers have concerns about the analysis of polynomial expressivity, which are not fully resolved in the rebuttal phase.

After discussing with SAC, we finally have a consensus about accepting this work.

**Justification For Why Not Higher Score:**

The proposed method is effective and efficient, and some interesting analysis is provided. However, the discussion of polynomial expressivity is too coarse at the current stage.

**Justification For Why Not Lower Score:**

The performance of the proposed method is good, especially on large-scale graphs.
The proposed method is easy to implement, and its rationality is analyzed (although the analysis is coarse).

---

### Decision · Program_Chairs · 2024-01-16

Accept (poster)